# Stochastic Multi-armed Bandits: Optimal Trade-off among Optimality, Consistency, and Tail Risk

**David Simchi-Levi**
Institute for Data, Systems, and Society
Massachusetts Institute of Technology
Cambridge, MA 02139
dslevi@mit.edu

**Zeyu Zheng**
Industrial Engineering & Operations Research
University of California, Berkeley
Berkeley, CA 94720
zyzheng@berkeley.edu

**Feng Zhu**
Institute for Data, Systems, and Society
Massachusetts Institute of Technology
Cambridge, MA 02139
fengzhu@mit.edu

## Abstract

We consider the stochastic multi-armed bandit problem and fully characterize the interplays among three desired properties for policy design: worst-case optimality, instance-dependent consistency, and light-tailed risk. We show how the order of expected regret exactly affects the decaying rate of the regret tail probability for both the worst-case and instance-dependent scenario. A novel policy is proposed to achieve the optimal regret tail risk for any regret threshold. Concretely, for any given $\alpha \in [1/2, 1)$ and $\beta \in [0, 1)$, our policy achieves a worst-case expected regret of $\tilde{O}(T^\alpha)$ and instance-dependent expected regret of $\tilde{O}(T^\beta)$, while enjoys a probability of incurring an $\Omega(T^\delta)$ regret that decays exponentially with a polynomial $T$ term. Such decaying rate is proved to be best achievable. We also generalize our analysis to the stochastic multi-armed bandit problem with non-stationary baseline rewards, where in each time period $t$, the decision maker pulls one of $K$ arms and collects a reward which is the sum of three terms: the mean of the pulled arm, an independent noise, and a non-stationary baseline reward as a function of $t$. Our results reveal insights on the trade-off between expected regret and tail risk for both worst-case and instance-dependent scenario, indicating that more sub-optimality and inconsistency leaves space for more light-tailed risk of incurring a large regret.

## 1 Introduction

The stochastic multi-armed bandit (MAB) problem is a well-established area of research in sequential decision-making under uncertainty, applications of which include online advertising, recommendation systems, digital clinical trials, etc. Let $T$ be a pre-fixed time horizon. In a standard MAB problem, the decision maker selects one of several arms (rewards of which are unknown a priori) in each time period $t \in [T]$ with the goal of maximizing the expected sum of rewards over all time periods. The MAB problem highlights the exploration-exploitation trade-off: the decision maker must balance between exploring arms with relatively unknown reward distributions and exploiting arms with relatively known high expected rewards. There is a significant amount of literature on MAB, with a comprehensive review provided in [25].

In order to evaluate policy performance and guide policy design, much of the MAB literature uses the metric of maximizing the *expected cumulative reward*, or equivalently minimizing the **expected**

37th Conference on Neural Information Processing Systems (NeurIPS 2023).

**regret** (where *regret* is generally defined as the difference between the cumulative reward of pulling the best arm and the cumulative reward of a policy). However, it has recently been documented that if an MAB policy design only focuses on minimizing the **expected regret**, the policy may be exposed to heavy-tailed risks of incurring a large regret; see [11] and [24] for detailed documentation. Roughly speaking, designing a policy that focuses only on the expected regret could be analogous to designing an investment portfolio that focuses only on the expected return without looking at other risks. As recently documented in [24], many renowned algorithms (e.g., Successive Elimination (SE) [10], Thompson Sampling (TS) [20]), despite of enjoying optimality on expected regret, can incur a "heavy-tailed risk" — the probability of incurring a linear regret slowly decays at a polynomial rate $\Omega(\text{poly}(1/T))$ as $T$ tends to infinity. In contrast, a "light-tailed" risk in this MAB setting means that the probability of a policy incurring a linear regret decays at an exponential rate $\exp(-\Omega(T^\gamma))$ for some $\gamma > 0$.

Two important and popular notions in the MAB literature to describe properties of a policy — worst-case *optimality* and instance-dependent *consistency* — are both defined in terms of expected regret. There lacks an understanding in the literature about how much regret tail risk will arise by pursuing optimality and/or consistency for policy design. In particular:

> ***How do different levels of optimality and consistency respectively affect the tail risk?***
> ***What is the optimal trade-off among optimality, consistency, and light-tailed risk?***

Motivated by these questions, we summarize our contributions in Section 1.1. To facilitate describing the results on regret orders and function orders, we use $O(\cdot)$ ($\tilde{O}(\cdot)$) and $\Omega(\cdot)$ ($\tilde{\Omega}(\cdot)$) to present upper and lower bounds on the growth rate up to constant (logarithmic) factors, respectively, and $\Theta(\cdot)$ ($\tilde{\Theta}(\cdot)$) to characterize the rate when the upper and lower bounds match up to constant (logarithmic) factors. We use $o(\cdot)$ and $\omega(\cdot)$ to present strictly dominating upper bounds and strictly dominated lower bounds, respectively.

## 1.1 Our Contributions

1. We characterize the connections and interplays among worst-case optimality, instance-dependent consistency, and light-tailed risk. We show that relaxing worst-case regret order or instance-dependent regret order can help make the regret tail lighter in an information-theoretic way. Given the family of policies with a worst-case regret of $\tilde{O}(T^\alpha)$ and instance-dependent regret of $\tilde{O}(T^\beta)$, we demonstrate how fast we can best hope the probability of incurring an $\Omega(T^\delta)$ decays with $T$, for any feasible tuple $(\alpha, \beta, \delta)$. (This can be viewed as a best-achievable "lower bound" result. The next item describes a matching "upper bound" result.)

2. We design a simple policy that, for any given $\alpha$ and $\beta$, obtains $\tilde{O}(T^\alpha)$ worst-case regret and $\tilde{O}(T^\beta)$ instance-dependent regret, and simultaneously obtains the best achievable regret tail probability for both worst-case and instance-dependent scenarios. Our regret tail bounds are explicit formulas showing that the probability of incurring a worst-case regret larger than any $x$ is bounded by $\exp(-\tilde{\Omega}((x/T^{1-\alpha}) \wedge T^\beta))$, and that the probability of incurring an instance-dependent regret larger than any $x$ is bounded by $\exp(-\tilde{\Omega}(T^\beta))$. Our policy design builds upon the idea of successive elimination, and constructs novel bonus terms to balance among worst-case optimality, instance-dependent consistency, and light-tailed risk.

3. Our theory is generalized to an advanced MAB model that allows a common non-stationary baseline among all arms for each time period. Different time periods can have different baseline rewards. The model may accommodate applications with structured non-stationarity beyond those from standard multi-armed bandit problems where arm rewards are assumed to be stationary. We apply a simple and novel technique of random permutation to deal with this additional feature. Last, a brief account of experiments are conducted to illustrate our theoretical findings. We discuss how tuning parameters affect the performance of our policy, and reiterate the insight that relaxing worst-case optimality and instance-dependent consistency (or allowing sub-optimality and inconsistency) may leave space for regret distribution being more light-tailed.

## 1.2 Related Work

Our work builds upon the stochastic multi-armed bandit (MAB) literature, and reviews can be found in [4, 20, 25, 17]. However, the tail risk of stochastic bandit algorithms is not well understood. [2, 21] showed that the policy regret generally concentrates only at a polynomial rate. That is, the probability of incurring $c(\ln T)^p$ regret (with $c > 0$ and $p > 1$ fixed) is polynomially decaying with $T$. [1] showed that a policy that obtains logarithmic regret can be sensitive to mis-specified risk parameter. That is, if one underestimated the risk parameter $\sigma$ for subGaussian noises, then one may obtain a large regret. Note that the goal here is still to minimize the expected regret.

Our work is inspired by [11, 24]. [11] first analyzed heavy-tailed risk for bandit algorithms and showed that the family of information-theoretically optimized bandit policies suffers from serious heavy-tailed risk: the probability of incurring a linear regret is at least $\Omega(1/T)$. They also modified UCB algorithms to reach a desired tail risk polynomially dependent on $T$, making the algorithms more robust to mis-specifications. [24] showed the incompatibility between instance-dependent consistency and light-tailed risk. They also proposed a simple policy design showing that worst-case optimality (expected regret bounded by $\tilde{O}(\sqrt{T})$) permits light-tailed risk (tail bound exponentially decaying with $\sqrt{T}$). However, the interplay among optimalilty, consistency, and tail risk remains unclear. This is the question we want to fully address in this work.

Our work is also related with multi-tasking bandits, where minimizing expected regret is not the only goal (see, e.g., [8, 9, 28, 29, 23]). In particular, [23] studied the optimal trade-off between efficiency (obtaining low regret) and statistical power (estimating arm gaps correctly). They also extended their analysis to the MAB problem with non-stationary baseline rewards. We note that their task is still centered on expectation, and their results are entirely based on an instance-dependent perspective. Another line of related work is risk-averse formulations of the stochastic MAB problem (see, e.g., [22, 12, 18, 31, 27, 5, 26, 19, 30, 3, 14, 7, 6]). These formulations consider different objectives other than expected regret, such as mean-variance and value-at-risk. Our work's objective and metric (levels of tail risks) are different, which leads to different policy design and analysis.

## 1.3 Organization and Notation

The rest of the paper is organized as follows. In Section 2, we discuss the setup and introduce the key concepts. In Section 3, we show our main results. In Section 4, we extend our main results. In Section 5, we conclude. Experiments are provided in the supplementary material. Before proceeding, we introduce some notation. For any $a, b \in \mathbb{R}$, $a \wedge b = \min\{a, b\}$, $a \vee b = \max\{a, b\}$, and $a_+ = \max\{a, 0\}$. We denote $[N] = \{1, \cdots, N\}$ for any positive integer $N$. Next, we state our problem setting and describe all the notation used in the problem setting.

## 2 The Setup

### 2.1 The Model

Fix $T$ as the time horizon and $K$ as the number of different arms. Without loss of generality, we presume that $K \geq 2$, $T \geq 3$, and $T \geq K$. In each time period $t \in [T]$, the decision maker (DM) has access to all prior information and then chooses an arm $a_t \in [K]$ to pull. Upon pulling the arm, the DM observes and receives a reward $r_{t,a_t}$. Specifically, let $H_t = \{a_1, r_{1,a_1}, \cdots, a_{t-1}, r_{t-1,a_{t-1}}\}$ be the history information prior to time $t$. $H_1$ is set as $\emptyset$ for definition completeness.

In this work we consider policies that are allowed to utilize the knowledge of $T$. When we say $\pi$ is policy, we mean that $\pi$ is composed of a series of "sub-policies": $\pi(1), \cdots, \pi(T), \cdots$. That is, with the prior knowledge of $T$, $\pi$ executes $\pi(T)$ throughout the whole time horizon. At time $t$, the action taken is determined by $\pi_t(T) : H_t \cup \{T\} \longmapsto a_t$, which can potentially be random. Different $T$'s may lead to completely different sub-policies.

The time-$t$ reward is independently sampled as $r_{t,a_t} = \theta_{a_t} + \epsilon_{t,a_t}$, where $\theta_{a_t}$ is the mean reward of arm $a_t$, and $\epsilon_{t,a_t}$ is a zero-mean random noise, independent across time periods. The random noise $\epsilon_{t,a_t}$ is assumed to be $\sigma$-subGaussian. That is, there exists a $\sigma > 0$ such that for any arm $k$ and time $t$,

$$\max\left\{\mathbb{P}\left(\epsilon_{t,k} \geq x\right), \mathbb{P}\left(\epsilon_{t,k} \leq -x\right)\right\} \leq \exp(-x^2/(2\sigma^2)).$$

Let $\theta = (\theta_1, \cdots, \theta_K)$ be the vector of means. Let $\theta_* = \max\{\theta_1, \cdots, \theta_K\}$ be the optimal mean reward among the $K$ arms. Note that the DM does not know $b_t$ or $\theta$ a priori, except that $b_t \in [0, B]$ (for some $B \geq 0$) and $\theta \in [0, 1]^K$. The empirical regret of the policy $\pi = (\pi_1, \cdots, \pi_T)$ under the mean vector $\theta$ and the noise parameter $\sigma$ over a time horizon of $T$ is defined as

$$\hat{R}_{\theta,\sigma}^{\pi(T)}(T) = \theta_* \cdot T - \sum_{t=1}^{T} (\theta_{a_t} + \epsilon_{t, a_t}).$$

Let $\Delta_k = \theta_* - \theta_k$ denote the gap between the optimal arm and the $k$th arm. Define $\Delta_0 > 0$ such that

$$1/\Delta_0 = \sum_{k' : \Delta_{k'} > 0} 1/\Delta_{k'}.$$

Denote $n_{t,k} = \sum_{s=1}^{t} \mathbb{1}\{a_s = k\}$ as the number of times arm $k$ has been pulled up to time $t$. We sometimes use $n_k = n_{T,k}$ to denote the total number of times arm $k$ pulled through the entire time horizon. We denote $t_k(n)$ as the time period that arm $k$ is pulled for the $n$-th time. We define the pseudo regret as

$$R_{\theta,\sigma}^{\pi(T)}(T) = \sum_{k=1}^{K} n_k \Delta_k$$

and the aggregated random noise as

$$N^\pi(T) = \sum_{t=1}^{T} \epsilon_{t, a_t} = \sum_{k=1}^{K} \sum_{m=1}^{n_k} \epsilon_{t_k(m), k}.$$

Therefore, the empirical regret can also be written as $\hat{R}_{\theta,\sigma}^\pi(T) = R_{\theta,\sigma}^\pi(T) - N^\pi(T)$. In this work, for notation simplicity, we will write $\hat{R}_\theta^\pi(T)$ instead of $\hat{R}_{\theta,\sigma}^{\pi(T)}(T)$ and $R_\theta^\pi(T)$ instead of $R_{\theta,\sigma}^{\pi(T)}(T)$. The following simple lemma gives the mean and the tail probability of the aggregated noise $N^\pi(T)$.

**Lemma 2.1.** *We have* $\mathbb{E}[N^\pi(T)] = 0$ *and*

$$\max\left\{\mathbb{P}\left(N^\pi(T) \geq x\right), \mathbb{P}\left(N^\pi(T) \leq -x\right)\right\} \leq \exp\left(\frac{-x^2}{2\sigma^2 T}\right).$$

In the worst-case scenario, $x$ is taken as $\Omega(\sqrt{T})$, and so the tail in Lemma 2.1 is negligible. In the instance-dependent scenario, $x$ may take $o(\sqrt{T})$, making the tail in Lemma 2.1 no longer ignorable. We note that the empirical regret unavoidably incurs a tail in Lemma 2.1 due to the appearance of genuine noise, even if we always pull the optimal arm. Therefore, we will mainly focus on pseudo regret in our subsequent discussions.

## 2.2 Worst-case Optimality, Instance-dependent Consistency, Light-tailed Risk

Now we describe concepts that are needed to formalize the policy design and analysis.

**1. Worst-case $\alpha$-optimality.** Fix $\alpha \in [1/2, 1)$. A policy is said to be worst-case $\alpha$-optimal or simply, *$\alpha$-optimal*, if for any $\varepsilon > 0$, the policy has that

$$\limsup_{T \to +\infty} \frac{\sup_\theta \mathbb{E}\left[R_\theta^\pi(T)\right]}{T^{\alpha + \varepsilon}} = 0.$$

In brief, a policy is $\alpha$-optimal if the worst-case expected regret can never be growing in $T$ at a polynomial rate faster than $T^\alpha$. Intuitively, the smaller the $\alpha$ is, the better performance a policy has in terms of worst-case expected regret order.

**2. Instance-dependent $\beta$-consistency.** Fix $\beta \in [0, 1)$. A policy is called instance-dependent $\beta$-consistent or simply $\beta$-consistent, if for any underlying true mean vector $\theta$ and any $\varepsilon > 0$, the policy has that

$$\limsup_{T \to +\infty} \frac{\mathbb{E}\left[R_\theta^\pi(T)\right]}{T^{\beta + \varepsilon}} = 0.$$

In brief, a policy is $\beta$-consistent if the expected regret can never grow faster than $T^\beta$ for any fixed instance. We note that the "worst-case" notion and the "instance-dependent" notion, in these two items, are most commonly used in the bandits literature, and both notions care about the **expectation** of the regret distribution. The next notion concerns the tail of regret distribution.

**3. $(\delta, \gamma)$-tail.** Fix $\delta \in (0, 1]$ and $\gamma \in [0, 1]$. We differentiate between two scenarios: worst-case and instance-dependent.

(a). A policy is called worst-case $(\delta, \gamma)$-tailed, if there exists constants $c \in (0, 1/2)$ and $C > 0$ such that

$$\limsup_{T \to +\infty} \frac{\ln \left\{ \sup_\theta \mathbb{P} \left( R_\theta^\pi(T) > cT^\delta \right) \right\}}{T^\gamma} \le -C.$$

Both $c$ and $C$ are allowed to rely on $(\delta, \gamma)$. More intuitively, the above definition means that a policy is worst-case $(\delta, \gamma)$-tailed if the worst-case probability of incurring a regret of $T^\delta$ can be bounded by an exponential term of polynomial $T^\gamma$:

$$\sup_\theta \mathbb{P} \left( R_\theta^\pi(T) > cT^\delta \right) = \exp(-\Omega(T^\gamma)).$$

All else equal, one would prefer a policy with larger $\gamma$ so that the tail probability of incurring a big regret is lighter.

(b). A policy is called instance-dependent $(\delta, \gamma)$-tailed, if for any underlying true mean vector $\theta$, there exists constants $c \in (0, 1/2)$ and $C > 0$ such that

$$\limsup_{T \to +\infty} \frac{\ln \left\{ \mathbb{P} \left( R_\theta^\pi(T) > cT^\delta \right) \right\}}{T^\gamma} \le -C.$$

Note that the constant $C$ may be dependent on both $c$ and $\theta$. In brief, a policy is instance-dependent $(\delta, \gamma)$-tailed if the instance-dependent probability of incurring a regret of $T^\delta$ can be bounded by an exponential term of polynomial $T^\gamma$:

$$\mathbb{P} \left( R_\theta^\pi(T) > cT^\delta \right) = \exp(-\Omega(T^\gamma)).$$

**Remarks.**

1. When defining the tail, we impose $c \in (0, 1/2)$ to avoid the corner case when $\delta = 1$. In such case, if $c \ge 1$, the tail probability is zero because $\theta \in [0, 1]^K$. We note that when $\delta < 1$, the condition $c \in (0, 1/2)$ is not essential, and here we retain it for simplicity of exposition.

2. It is well known that for the stochastic MAB problem, one can design algorithms to achieve both 0-consistency and 1/2-optimality. Among them, two types of policies are of prominent interest: Successive Elimination (SE) and Upper Confidence Bound (UCB). The bonus term (or, the confidence radius) $\text{rad}(n)$ is typically set as

$$\text{rad}(n) = \sigma \sqrt{\frac{\eta \ln T}{n}} \tag{1}$$

with $\eta > 0$ being some tuning parameter. That said, both the SE policy and UCB policy may not perform well in terms of tail probability of incurring a large regret, as documented in [11] and [24].

## 3 Main Results

### 3.1 Tail Lower Bound: The Best to Hope

In this section, we show how fast the regret tail can decay as a function of $T$, given that a policy is $\alpha$-optimal or/and $\beta$-consistent. More concretely, if a policy is $\alpha$-optimal or/and $\beta$-consistent, Theorem 3.1 shows what is the fastest decaying rate we can hope for the probability that the pseudo regret is at least $\Omega(T^\delta)$ (in either the worst-case or instance-dependent scenario).

**Theorem 3.1.** *We have the following arguments.*

*1. Fix $\alpha \in [1/2, 1)$. If a policy $\pi$ is $\alpha$-optimal, then for any $\delta > \alpha$ and $\gamma > \delta + \alpha - 1$, $\pi$ can never be worst-case $(\delta, \gamma)$-tailed.*

*2. Fix $\beta \in [0, 1)$. If a policy $\pi$ is $\beta$-consistent, then for any $\delta > \beta$ and $\gamma > \beta$, $\pi$ can never be instance-dependent $(\delta, \gamma)$-tailed.*

In the proof of Theorem 3.1, briefly, for the worst-case scenario, we construct a series of instance pairs such that the gap between two arms shrink with a polynomial rate of $1/T$. For the instance-dependent scenario, we fix a pair of instances and investigates how the tail probability scales in the two environments as $T$ increases. An intermediate step towards doing the complete proof is a lemma showing that if the policy is "effective", i.e., achieves a sub-linear regret under either case, then the estimation of the *sub-optimal* arm becomes more precise in probability as $T$ increases. Details are provided in the supplementary material.

Theorem 3.1 implies Corollary 3.2, which is a direct application of the following argument — if a policy is *not* instance-dependent $(\delta, \gamma)$-tailed, then it is also not worst-case $(\delta, \gamma)$-tailed.

**Corollary 3.2.** *Fix $\alpha \in [1/2, 1)$ and $\beta \in [0, 1)$. If a policy $\pi$ is both $\alpha$-optimal and $\beta$-consistent, then we have the following arguments.*

*1. For any $\delta > \alpha$ and $\gamma > (\delta + \alpha - 1) \wedge \beta$, $\pi$ can never be worst-case $(\delta, \gamma)$-tailed.*

*2. For any $\delta > \beta$ and $\gamma > \beta$, $\pi$ can never be instance-dependent $(\delta, \gamma)$-tailed.*

To help better understand Corollary 3.2 in an intuitive way, we let $x = \Theta(T^\delta)$, then the critical values of $T^\gamma$ for different scenarios and cases are listed in Table 1. That said, the best we can hope for the order of the regret tail bounds cannot decay faster than the critical values.

| $\ln \sup_\theta \mathbb{P}_\theta^\pi(\text{Regret} > x)$ (worst-case scenario) | $-(x/T^{1-\alpha}) \wedge T^\beta$ for large $x$ |
|---|---|
| $\ln \mathbb{P}_\theta^\pi(\text{Regret} > x)$ (instance-dependent scenario) | $-T^\beta$ for large $x$ |

Table 1: Citical values of log tail probability for policies that are both $\alpha$-optimal and $\beta$-consistent

## 3.2 Tail Upper Bound: The Best to Achieve

In this section, we show that "the best we can hope" is achievable by concrete policies. Given $\alpha$ and $\beta$, our goal is to design a policy such that the decaying rate of the log tail probabilities match the critical values indicated in Table 1. Namely, we design a *single* policy that is $\alpha$-optimal and $\beta$-consistent, and enjoys the optimal tail rate in *both* worst-case and instance-dependent scenarios.

---
**Algorithm 1** Successive Elimination with Random Permutation (SEwRP)
---
$\mathcal{A}_1 = [K]$. $t \leftarrow 0$. $i \leftarrow 1$.
**while** $t < T$ **do**
    Sample a permutation $\pi_i$ of $\mathcal{A}_i$ uniformly at random.
    Pull each arm in $\mathcal{A}_i$ once sequentially according to $\pi_i$.
    $t \leftarrow t + |\mathcal{A}_i|$.
    $\mathcal{B}_i = \{k | \exists k' : \hat{\mu}_{t,k'} - \text{rad}(n_{t,k'}) > \hat{\mu}_{t,k} + \text{rad}(n_{t,k})\}$. $\mathcal{A}_{i+1} = \mathcal{A}_i \setminus \mathcal{B}_i$.
    $i \leftarrow i + 1$.
**end while**
---

Algorithm 1 is designed upon the standard Successive Elimination (SE) algorithm. First, for each phase $i$, we add a random permutation step prior to taking the actions. This randomization step will be useful for extending our theories. Second, the bonus term $\text{rad}(n)$ in SEwRP is different from standard bonus terms like (1) or those used in [24] — we need to reach balance among optimality, consistency, and tail risk. The concrete design is given in Theorem 3.3, which fully characterizes the regret tail bound of Algorithm 1 in both worst-case and instance-dependent scenarios.

**Theorem 3.3.** *For the $K$-armed bandit problem, $\pi = \text{SEwRP}$ with*

$$\text{rad}(n) = \eta_1 \frac{(T/K)^\alpha \sqrt{\ln T}}{n} \wedge \eta_2 \sqrt{\frac{T^\beta \ln T}{n}} \tag{2}$$

*satisfies the following properties: fix any $\eta_1, \eta_2 \geq 0$, for any $x > 0$, we have*

*1. (worst-case regret tail)*

$$\sup_{\theta} \mathbb{P}(R_{\theta}^{\pi}(T) \geq x) \leq 6K \exp\left(-\frac{\left(x - K - 4\eta_1 K^{1-\alpha} T^{\alpha}\sqrt{\ln T}\right)_+^2}{32\sigma^2 KT}\right)$$

$$+ 6K^2 T \exp\left(-\frac{\sqrt{\ln T}}{8\sigma^2}\left(\frac{\eta_1(x-K)_+}{2K^{\alpha}T^{1-\alpha}} \wedge \eta_2^2 T^{\beta}\sqrt{\ln T}\right)\right). \qquad (3)$$

*2. (instance-dependent regret tail)*

$$\mathbb{P}(R_{\theta}^{\pi}(T) \geq x) \leq 3K \exp\left(-\frac{\left((x-K)\Delta_0 - 4\eta_2^2 T^{\beta}\ln T\right)_+}{8\sigma^2}\right)$$

$$+ 3KT \sum_{k:\Delta_k>0} \exp\left(-\frac{\left(\eta_1(T/K)^{\alpha}\Delta_k \wedge \eta_2^2 T^{\beta}\sqrt{\ln T}\right)\sqrt{\ln T}}{8\sigma^2}\right), \qquad (4)$$

The design of our bonus term is novel and hopefully provides additional insights, as follows. The first component can be interpreted as controlling the worst-case tail risk, while the second one can be regarded as controlling the instance-dependent tail risk. There exhibits a *phase transition* with respect to the size of confidence interval. At the beginning $\tilde{\Theta}(T^{2\alpha-\beta})$ time periods, the second term dominates, and so the confidence interval shrinks at a rate of $1/\sqrt{n}$, suggesting that we focus more on exploration within the consistency constraint. While in the remaining time periods, the first term dominates, and so the confidence interval shrinks at a rate of $1/n$, suggesting that we focus more on exploitation within the optimality condition. Our policy design suggests that in real-world practice, to achieve more light-tailed risk, it might be beneficial to have two different phases in the policy design: more exploration at the beginning, and more exploitation afterwards.

The following proposition shows that our policy has $\tilde{O}(T^{\alpha})$ worst-case expected regret and $\tilde{O}(T^{\beta})$ instance-dependent expected regret.

**Proposition 3.4.** *For the $K$-armed bandit problem, $\pi =$ SEwRP with (2) satisfies the following expected regret bounds (ignoring additive and multiplicative constant terms). If $\beta > 0$ or $\beta = 0$ and $\eta_2 = \Omega(1 \vee \sigma)$, then*

$$\sup_{\theta} \mathbb{E}\left[R_{\theta}^{\pi}(T)\right] = O\left(K^{1-\alpha} T^{\alpha}\sqrt{\ln T}\right) \quad and \quad \mathbb{E}\left[R_{\theta}^{\pi}(T)\right] = O\left(T^{\beta}\ln T/\Delta_0\right).$$

**Proof road-map.** Without loss of generality, we assume arm 1 is optimal. Let the regret threshold $x \geq K$. For any $k \neq 1$, we define

$$S_k = \{\text{Arm 1 is not eliminated before arm } k\}.$$

In the following, we differentiate between two scenarios.

For the worst-case scenario, we define

$$\mathcal{A}^* = \{k \neq 1 : n_k \leq 1 + T/K\}.$$

Then with some basic algebra, one can build

$$\mathbb{P}\left(R_{\theta}^{\pi}(T) \geq x\right) \leq \sum_{k\neq 1} \mathbb{P}\left((n_k - 1)\Delta_k \geq \frac{x-K}{2K}, \; k \in \mathcal{A}^*\right) + \sum_{k\neq 1} \mathbb{P}\left(\Delta_k \geq \frac{x-K}{2T}, \; k \notin \mathcal{A}^*\right) \qquad (5)$$

We need to emphasize that introducing $\mathcal{A}^*$ allows us to establish $\Delta_k = \Omega(x/T)$, otherwise we may only have $\Delta_k = \Omega(x/KT)$. The goal of defining $\mathcal{A}^*$ is to make sure our analysis aligns with our new bonus design (we have a $1/K^{\alpha}$ factor in the numerator) such that our tail probabilities have optimal dependence on the number of arms $K$.

We decompose the tail risk of an algorithm by two types of events: (i) spending too much time before correctly discarding a sub-optimal arm; (ii) wrongly discarding the optimal arm.

1. Consider the case when $S_k$ happens. This corresponds to the tail risk of *spending too much time before correctly discarding a sub-optimal arm* [24]. For either $k \in \mathcal{A}^*$ or $k \notin \mathcal{A}^*$, we fix some non-random number $n_0 < n_k$ (the choice of $n_0$ is important: to be precise, we let $n_0 = \lceil (x - K)/(2K\Delta_k) \rceil$ when $j \in \mathcal{A}^*$ and $n_0 = T/K$ when $j \notin \mathcal{A}^*$) where $n_0$ is a phase when a sub-optimal arm $k$ is not eliminated by the optimal arm 1. Then arm 1 and $k$ are both not eliminated after each of them being pulled $n_0$ times, which means

$$\hat{\mu}_{t_1(n_0),1} - \mathrm{rad}(n_0) \leq \hat{\mu}_{t_k(n_0),k} + \mathrm{rad}(n_0).$$

The probability of this event can be bounded using concentration properties of sub-Gaussian random variables.

2. Now consider the case when $\bar{S}_k$ happens. This corresponds to the tail risk of *wrongly discarding the optimal arm* [11, 24]. Then after some phase $n$, the optimal arm 1 is eliminated by some arm $k'$, while arm $k$ is not eliminated. We emphasize that $k = k'$ does not necessarily hold when $K > 2$. As a result, the following two events hold simultaneously:

$$\hat{\mu}_{t_{k'}(n),k'} - \mathrm{rad}(n) \geq \hat{\mu}_{t_1(n),1} + \mathrm{rad}(n) \quad \text{and} \quad \hat{\mu}_{t_k(n),k} + \mathrm{rad}(n) \geq \hat{\mu}_{t_1(n),1} + \mathrm{rad}(n).$$

The two events leads to, respectively,

$$\text{mean of } n \text{ noise terms} \geq \mathrm{rad}(n) = \tilde{\Theta}\left( \frac{(T/K)^\alpha}{n} \wedge \sqrt{\frac{T^\beta}{n}} \right)$$

and

$$\text{mean of } n \text{ noise terms} \geq \Delta_k = \Omega\left( \frac{x}{T} \right).$$

Then the tail probability can be further bounded by

$$O\left( \exp\left( -\left( \frac{(T/K)^{2\alpha}}{n} \vee T^\beta \right) \wedge \frac{nx^2}{T^2} \right) \right) = O\left( \exp\left( -\frac{x}{K^\alpha T^{1-\alpha}} \wedge T^\beta \right) \right).$$

which yields the second term in (3).

For the instance-dependent scenario, the proof follows similarly to that for the worst-case scenario, but with some amendment. For each $k \neq 1$, we treat $\Delta_k$ as a constant, and instead of (5), we use

$$\mathbb{P}\left( R_\theta^\pi(T) \geq x \right)$$

$$\leq \sum_{k:\Delta_k>0} \mathbb{P}\left( (n_k - 1)\Delta_k \geq \frac{(x - K)/\Delta_k}{\sum_{k':\Delta_{k'}>0} 1/\Delta_{k'}}, S_k \right)$$

$$+ \sum_{k:\Delta_k>0} \mathbb{P}\left( (n_k - 1)\Delta_k \geq \frac{(x - K)/\Delta_k}{\sum_{k':\Delta_{k'}>0} 1/\Delta_{k'}}, \bar{S}_k \right).$$

Here, we do not need $\mathcal{A}^*$ to control the dependence on $K$. Also, when bounding the probability when $S_k$ happens, we accordingly define

$$n_0 = \lceil (x - K)/\Delta_k^2 \cdot \Delta_0 \rceil \leq n_k - 1.$$

The detailed proof is left to the supplementary material. We would like to emphasize the technical novelty compared to that in [24]. In general, since $\alpha$ and $\beta$ become flexible constants, the proof requires more delicate controls on probability bounds. When proving the worst-case upper bound, we need a careful manipulation on $\mathrm{rad}(n)$ since we are dealing with the minimum of two different types of bonus terms. When proving the instance-dependent upper bound, we require a careful division of the tail event to make the bound as tight as possible, depending on specific instances ($\theta$). In [24], the tail bound is only concerned with the worst-case scenario with $\alpha = 1/2$, and hence the aforementioned challenges do not exist.

Finally, we would like to note that although applying the naive explore-then-commit (ETC) strategy might be able to circumvent heavy-tailed risk, the approach has two main issues:

1. From the regret expectation perspective, the ETC policy can only achieve worst-case $O(T^{2/3})$ expected regret bound. When $\alpha \in [1/2, 2/3)$, we still have to consider other types of policies. Further, it seems unlikely that without knowing the arm gaps $\{\Delta_k\}$, the explore-then-commit policy can achieve $\tilde{O}(T^\alpha)$ worst-case regret and $\tilde{O}(T^\beta)$ instance-dependent regret simultaneously with $\beta < \alpha$.

2. From the regret tail risk perspective, the tail probability of incurring a large regret may not be optimal in the worst-case. Consider a simple 2-armed bandit case with arm gap $\Delta$. An ETC policy with $m = \Theta(T^\alpha)$ ($\alpha \in [2/3, 1)$) steps of exploration has a probability of $\Omega(\exp(-m\Delta^2))$ committing to a wrong arm. In the worst-case scenario, let $\Delta = T^{(\alpha-1)/2}$ and the regret threshold be $x = T^{(\alpha+1)/2}/4 \in (m\Delta, (T-m)\Delta)$. Then if we incur a regret of $x$, it means we commit to a wrong arm after the exploration, which suggests $\mathbb{P}(R_\theta^\pi(T) > x) = \Omega(\exp(-m\Delta^2)) = \Omega(\exp(-T^{2\alpha-1})) = \omega(\exp(-x/T^{1-\alpha}))$.

## 4 Extension with Structured Non-stationarity

In this section, we extend our results to a more general stochastic MAB model. In certain applications, the assumption that the rewards for one arm are i.i.d. generated may fail to hold. On the other hand, the non-stationary MAB models are at the other extreme being a bit too conservative, where no structural properties exist. The stochastic MAB model with non-stationary baseline rewards tries to strike a balance between stationary stochastic MAB and fully adversarial MAB. Such a structure is receiving attentions in the literature (see, e.g., [13, 16, 15, 23]). We restate the setting as follows and highlights those different from the standard stochastic MAB model introduced in Section 2.

At each time $t$, upon pulling an arm $a_t$, a reward is independently sampled as $r_{t,a_t} = b_t + \theta_{a_t} + \epsilon_{t,a_t}$, where $b_t$ is the baseline reward chosen by the nature at time $t$, $\theta_{a_t}$ is the mean reward of arm $a_t$, and $\epsilon_{t,a_t}$ is a zero-mean random noise, independent across time periods. The random noise $\epsilon_{t,a_t}$ is assumed to be $\sigma$-subGaussian. Let $\theta = (\theta_1, \cdots, \theta_K)$ be the vector of means and $\theta_* = \max\{\theta_1, \cdots, \theta_K\}$ be the optimal mean reward among the $K$ arms. Note that the DM does not know $b_t$ or $\theta$ a priori, except that $b_t \in [0, B]$ (for some $B \geq 0$) and $\theta \in [0, 1]^K$. The empirical regret of the policy $\pi = (\pi_1, \cdots, \pi_T)$ under the mean vector $\theta$ and the noise parameter $\sigma$ over a time horizon of $T$ is defined as

$$\hat{R}_\theta^\pi(T) = \theta_* \cdot T + \sum_{t=1}^T b_t - \sum_{t=1}^T (b_t + \theta_{a_t} + \epsilon_{t,a_t}) = \theta_* \cdot T - \sum_{t=1}^T (\theta_{a_t} + \epsilon_{t,a_t}).$$

Note that here although $b_t$ appears in the model, the term get cancelled in the regret. Let $\Delta_k = \theta_* - \theta_k$ denote the gap between the optimal arm and the $k$th arm. Same as Section 2.1, $\Delta_0$, the pseudo regret and the aggregated random noise are defined accordingly.

**Theorem 4.1.** *For the $K$-armed bandit problem with baseline rewards, $\pi$ = SEwRP with (2) satisfies the following properties: fix any $\eta_1, \eta_2 \geq 0$, for any $x > 0$, we have*

*1. (worst-case regret tail)*

$$\sup_\theta \mathbb{P}(R_\theta^\pi(T) \geq x) \leq 6K \exp\left(-\frac{\left(x - K - 4\eta_1 K^{1-\alpha} T^\alpha \sqrt{\ln T}\right)_+^2}{8(B+2\sigma)^2 KT}\right)$$

$$+ 6K^2 T \exp\left(-\frac{\sqrt{\ln T}}{2(B+2\sigma)^2}\left(\frac{\eta_1(x-K)_+}{2K^\alpha T^{1-\alpha}} \wedge \eta_2^2 T^\beta \sqrt{\ln T}\right)\right). \quad (6)$$

*2. (instance-dependent regret tail)*

$$\mathbb{P}(R_\theta^\pi(T) \geq x) \leq 3K \exp\left(-\frac{\left((x-K)\Delta_0 - 4\eta_2^2 T^\beta \ln T\right)_+}{2(B+2\sigma)^2}\right)$$

$$+ 3KT \sum_{k:\Delta_k>0} \exp\left(-\frac{\left(\eta_1(T/K)^\alpha \Delta_k \wedge \eta_2^2 T^\beta \sqrt{\ln T}\right)\sqrt{\ln T}}{2(B+2\sigma)^2}\right), \quad (7)$$

**Proposition 4.2.** *For the $K$-armed bandit problem with baseline rewards, $\pi$ = SEwRP with (2) satisfies the following expected regret bounds (ignoring additive and multiplicative constant terms). If $\beta > 0$ or $\beta = 0$ and $\eta_2 = \Omega(1 \vee \sigma)$, then*

$$\sup_\theta \mathbb{E}[R_\theta^\pi(T)] = O\left(K^{1-\alpha} T^\alpha \sqrt{\ln T}\right) \quad \text{and} \quad \mathbb{E}[R_\theta^\pi(T)] = O\left(T^\beta \ln T/\Delta_0\right).$$

The proof of Theorem 4.1 follows that of Theorem 3.3, except that when bounding the mean of error terms, we now have not only the true noise $\epsilon$, but also a baseline $b_t$. Lemma 4.3 is established to characterize the concentration of these combined error terms. We would like to note that the randomization step in SEwRP is crucial to hedge against the baseline rewards $b_t$, for which deterministic algorithms may fail. With uniform permutation, although the estimation of one arm is still biased, the difference of estimations between two different arms becomes unbiased.

**Lemma 4.3.** *Let $\tau_i$ be the (random) time period that we pull the last arm in $\mathcal{A}_i$. For any $n \geq 1$ and $k, k' \in [K]$, define*

$$E_n(k, k') = \frac{1}{n} \sum_{t=1}^{\tau_n \wedge T} (b_t + \epsilon_{t,a_t})(\mathbb{1}\{a_t = k\} - \mathbb{1}\{a_t = k'\}).$$

*We have, for any $x > 0$,*

$$\max\{\mathbb{P}\left(E_n(k, k') \geq x; \tau_n \leq T; k, k' \in \mathcal{A}_n\right),$$
$$\mathbb{P}\left(E_n(k, k') \leq -x; \tau_n \leq T; k, k' \in \mathcal{A}_n\right)\}$$
$$\leq 3 \exp\left(-\frac{nx^2}{2(B + 2\sigma)^2}\right).$$

## 5  Conclusion

In this work, we study the stochastic MAB problem with the goal of obtaining light-tailed regret distribution under given degree of worst-case optimality and instant-dependent consistency. We fully characterize the tradeoff among optimality, consistency, and tail risk via (i) proving lower bounds for decaying rate of incurring a large regret and (ii) proposing a novel algorithm with explicit regret tail upper bounds that match the lower bounds. We extend our analysis to the stochastic MAB model with non-stationary baseline rewards, and obtain similarly tight regret tail bounds. We study the empirical performance via synthetic simulations, and present insights of our results.

An open question for this work is whether our results can be generalized to the *any-time* case where the policy has no knowledge of $T$. For the lower bound, it has been pointed out in [21] that not knowing $T$ may have a significant impact on the concentration of regret distribution for standard UCB policies. For the upper bound, as is pointed out in [24], an SE type policy will fail to obtain light-tailed regret distribution under the any-time design even for the basic MAB model. Meanwhile, it seems nontrivial to incorporate randomization into confidence bound based policies to handle the stochastic MAB model with non-stationary baseline rewards. We leave the question for future work.

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
