# A Proofs for Section 3.1

**Lemma A.1.** *Consider the two-armed bandit problem with $\sigma$-Gaussian noise. We have the following arguments.*

*1. Let $\pi$ be a policy such that*

$$\limsup_{T \to +\infty} \frac{\sup_\theta \mathbb{E}\left[R_\theta^\pi(T)\right]}{T} = 0.$$

*That is, the worst-case expected regret under $\pi$ is sub-linear in $T$. Then for any $\varepsilon > 0$ and $\omega \in [0, 1/2)$, we have*

$$\limsup_{T \to +\infty} \sup_{\tilde{\theta}:1/2 \geq \tilde{\theta}_1 > \tilde{\theta}_2} \mathbb{P}_{\tilde{\theta}}^\pi(|\hat{\mu}_{T,2} - \tilde{\theta}_2| > \varepsilon/n_{T,2}^\omega) = 0.$$

*2. Let $\pi$ be a policy such that for any true mean vector $\theta$,*

$$\limsup_{T \to +\infty} \frac{\mathbb{E}\left[R_\theta^\pi(T)\right]}{T} = 0.$$

*That is, the instance-dependent expected regret under $\pi$ is always sub-linear in $T$. Then for any $\tilde{\theta} = (\tilde{\theta}_1, \tilde{\theta}_2)$ where $\tilde{\theta}_1 > \tilde{\theta}_2$, and any $\varepsilon > 0$, we have*

$$\limsup_{T \to +\infty} \mathbb{P}_{\tilde{\theta}}^\pi(|\hat{\mu}_{T,2} - \tilde{\theta}_2| > \varepsilon) = 0.$$

**Proof of Lemma A.1.**

1. Define

$$E_T = \left\{ |\hat{\mu}_{T,2} - \tilde{\theta}_2| \leq \varepsilon/n_{T,2}^\omega \right\}.$$

Fix any positive integer $N$, we have

$$\mathbb{P}_{\tilde{\theta}}^\pi(\bar{E}_T)$$
$$= \mathbb{P}_{\tilde{\theta}}^\pi(\bar{E}_T; n_{T,2} < N) + \mathbb{P}_{\tilde{\theta}}^\pi(\bar{E}_T; n_{T,2} \geq N)$$
$$\leq \mathbb{P}_{\tilde{\theta}}^\pi(n_{T,2} < N) + \sum_{n=N}^{+\infty} \mathbb{P}_{\tilde{\theta}}^\pi(\bar{E}_T; n_{T,2} = n)$$
$$\leq \mathbb{P}_{\tilde{\theta}}^\pi(n_{T,2} < N) + \sum_{n=N}^{+\infty} 2\exp(-\frac{n^{1-2\omega}\varepsilon^2}{2\sigma^2}).$$

Thus,

$$\limsup_T \sup_{\tilde{\theta}:1/2 \geq \tilde{\theta}_1 > \tilde{\theta}_2} \mathbb{P}_{\tilde{\theta}}^\pi(\bar{E}_T)$$

$$\leq \limsup_T \sup_{\tilde{\theta}:1/2 \geq \tilde{\theta}_1 > \tilde{\theta}_2} \mathbb{P}_{\tilde{\theta}}^\pi(n_{T,2} < N) + \sum_{n=N}^{+\infty} 2\exp(-\frac{n^{1-2\omega}\varepsilon^2}{2\sigma^2})$$

holds for any $N$. Note that the last term converges to 0 as $N \to +\infty$. It suffices to show

$$\sup_{\tilde{\theta}:1/2 \geq \tilde{\theta}_1 > \tilde{\theta}_2} \mathbb{P}_{\tilde{\theta}}^\pi(n_{T,2} < N) \to 0$$

as $T \to +\infty$ for any fixed $N$. Suppose this does not hold, then we can find $p > 0$, a sequence of times $\{T(m)\}_{m=1}^{+\infty}$ and a sequence of vectors $\{\tilde{\theta}(m)\}_{m=1}^{+\infty}$ (with $1/2 \geq \tilde{\theta}(m)_1 > \tilde{\theta}(m)_2$) such that

$$\mathbb{P}_{\tilde{\theta}(m)}^\pi(n_{T(m),2} < N) > p.$$

Let $M$ be some large number such that $q \triangleq p - N\exp(-\frac{M^2}{2\sigma^2}) > 0$. For each $m$, consider an alternative environment $\theta(m) = (\theta(m)_1, \theta(m)_2)$ where $\theta(m)_2 > \theta(m)_1 = \tilde{\theta}(m)_1$. Using the change of measure argument, we have

$$\mathbb{P}_{\theta(m)}^\pi(n_{T(m),2} < N)$$

$$= \mathbb{E}_{\theta(m)}^{\pi}[\mathbb{1}\{n_{T(m),2} < N\}]$$

$$= \mathbb{E}_{\tilde{\theta}(m)}^{\pi}\left[\exp\left(\sum_{n=1}^{n_{T(m),2}} \frac{(X_{t_2(n),2} - \tilde{\theta}(m)_2)^2 - (X_{t_2(n),2} - \theta(m)_2)^2}{2\sigma^2}\right)\mathbb{1}\{n_{T(m),2} < N\}\right]$$

$$= \mathbb{E}_{\tilde{\theta}(m)}^{\pi}\left[\exp\left(n_{T(m),2}\left(\frac{\tilde{\theta}(m)_2^2 - \theta(m)_2^2}{2\sigma^2} + \frac{(\theta(m)_2 - \tilde{\theta}(m)_2)\hat{\theta}_{T(m),2}}{\sigma^2}\right)\right)\mathbb{1}\{n_{T(m),2} < N\}\right]$$

$$\geq \mathbb{E}_{\tilde{\theta}(m)}^{\pi}\left[\exp\left(n_{T(m),2}\left(\frac{\tilde{\theta}(m)_2^2 - \theta(m)_2^2}{2\sigma^2} + \frac{(\theta(m)_2 - \tilde{\theta}(m)_2)\hat{\theta}_{T(m),2}}{\sigma^2}\right)\right)\cdot\right.$$
$$\left.\mathbb{1}\{\hat{\theta}_{T(m),2} > \tilde{\theta}(m)_2 - M, n_{T(m),2} < N\}\right]$$

$$\geq \mathbb{E}_{\tilde{\theta}(m)}^{\pi}\left[\exp\left(N\left(-\frac{(\tilde{\theta}(m)_2 - \theta(m)_2)^2}{2\sigma^2} - \frac{M(\theta(m)_2 - \tilde{\theta}(m)_2)}{\sigma^2}\right)\right)\cdot\right.$$
$$\left.\mathbb{1}\{\hat{\theta}_{T(m),2} > \tilde{\theta}_2 - M, n_{T(m),2} < N\}\right]$$

$$= \exp\left(N\left(-\frac{(\tilde{\theta}(m)_2 - \theta(m)_2)^2}{2\sigma^2} - \frac{M(\theta(m)_2 - \tilde{\theta}(m)_2)}{\sigma^2}\right)\right)\cdot$$
$$\mathbb{P}_{\tilde{\theta}}^{\pi}(\hat{\theta}_{T(m),2} > \tilde{\theta}(m)_2 - M, n_{T(m),2} < N)$$

$$\geq \exp\left(N\left(-\frac{1}{2\sigma^2} - \frac{M}{\sigma^2}\right)\right)\mathbb{P}_{\tilde{\theta}}^{\pi}(\hat{\theta}_{T(m),2} > \tilde{\theta}(m)_2 - M, n_{T(m),2} < N)$$

Note that

$$\mathbb{P}_{\tilde{\theta}}^{\pi}(\hat{\theta}_{T(m),2} > \tilde{\theta}(m)_2 - M, n_{T(m),2} < N)$$

$$> p - \sum_{n=1}^{N-1}\mathbb{P}_{\tilde{\theta}}^{\pi}(\hat{\theta}_{T(m),2} \leq \tilde{\theta}(m)_2 - M, n_{T(m),2} = n)$$

$$\geq p - \sum_{n=1}^{N-1}\exp(-\frac{nM^2}{2\sigma^2}) \geq p - N\exp(-\frac{M^2}{2\sigma^2}) = q > 0.$$

Therefore, there exists a constant positive probability such that $\pi$ pulls arm 2 no more than $N$ times under $\theta(m)$. As a result, $\pi$ incurs a worst-case linear expected regret, leading to a contradiction.

2. Define

$$E_T = \left\{|\hat{\mu}_{T,2} - \tilde{\theta}_2| \leq \varepsilon\right\}.$$

Fix any positive integer $N$, we have

$$\mathbb{P}_{\tilde{\theta}}^{\pi}(\bar{E}_T)$$

$$= \mathbb{P}_{\tilde{\theta}}^{\pi}(\bar{E}_T; n_{T,2} < N) + \mathbb{P}_{\tilde{\theta}}^{\pi}(\bar{E}_T; n_{T,2} \geq N)$$

$$\leq \mathbb{P}_{\tilde{\theta}}^{\pi}(n_{T,2} < N) + \sum_{n=N}^{+\infty}\mathbb{P}_{\tilde{\theta}}^{\pi}(\bar{E}_T; n_{T,2} = n)$$

$$\leq \mathbb{P}_{\tilde{\theta}}^{\pi}(n_{T,2} < N) + \sum_{n=N}^{+\infty}2\exp(-\frac{n\varepsilon^2}{2\sigma^2}).$$

Thus,

$$\limsup_{T}\mathbb{P}_{\tilde{\theta}}^{\pi}(\bar{E}_T)$$

$$\leq \limsup_{T}\mathbb{P}_{\tilde{\theta}}^{\pi}(n_{T,2} < N) + \sum_{n=N}^{+\infty}2\exp(-\frac{n\varepsilon^2}{2\sigma^2})$$

holds for any $N$. Note that the last term converges to $0$ as $N \to +\infty$. It suffices to show $\mathbb{P}^\pi_{\tilde\theta}(n_{T,2} < N) \to 0$ as $T \to +\infty$ for any fixed $N$. Suppose this does not hold, then we can find $p > 0$ and a sequence $\{T(m)\}^{+\infty}_{m=1}$ such that

$$\mathbb{P}^\pi_{\tilde\theta}(n_{T(m),2} < N) > p.$$

Let $M$ be some large number such that $q \triangleq p - N\exp(-\frac{M^2}{2\sigma^2}) > 0$. Consider an alternative environment $\theta = (\theta_1, \theta_2)$ where $\theta_2 > \theta_1 = \tilde\theta_1$. Using the change of measure argument, we have

$$\mathbb{P}^\pi_\theta(n_{T(m),2} < N)$$
$$= \mathbb{E}^\pi_\theta[\mathbb{1}\{n_{T(m),2} < N\}]$$
$$= \mathbb{E}^\pi_{\tilde\theta}\left[\exp\left(\sum_{n=1}^{n_{T(m),2}} \frac{(X_{t_2(n),2} - \tilde\theta_2)^2 - (X_{t_2(n),2} - \theta_2)^2}{2\sigma^2}\right)\mathbb{1}\{n_{T(m),2} < N\}\right]$$
$$= \mathbb{E}^\pi_{\tilde\theta}\left[\exp\left(n_{T(m),2}\left(\frac{\tilde\theta_2^2 - \theta_2^2}{2\sigma^2} + \frac{(\theta_2 - \tilde\theta_2)\hat\theta_{T(m),2}}{\sigma^2}\right)\right)\mathbb{1}\{n_{T(m),2} < N\}\right]$$
$$\geq \mathbb{E}^\pi_{\tilde\theta}\left[\exp\left(n_{T(m),2}\left(\frac{\tilde\theta_2^2 - \theta_2^2}{2\sigma^2} + \frac{(\theta_2 - \tilde\theta_2)\hat\theta_{T(m),2}}{\sigma^2}\right)\right)\mathbb{1}\{\hat\theta_{T(m),2} > \tilde\theta_2 - M, n_{T(m),2} < N\}\right]$$
$$\geq \mathbb{E}^\pi_{\tilde\theta}\left[\exp\left(N\left(-\frac{(\tilde\theta_2 - \theta_2)^2}{2\sigma^2} - \frac{M(\theta_2 - \tilde\theta_2)}{\sigma^2}\right)\right)\mathbb{1}\{\hat\theta_{T(m),2} > \tilde\theta_2 - M, n_{T(m),2} < N\}\right]$$
$$= \exp\left(N\left(-\frac{(\tilde\theta_2 - \theta_2)^2}{2\sigma^2} - \frac{M(\theta_2 - \tilde\theta_2)}{\sigma^2}\right)\right)\mathbb{P}^\pi_{\tilde\theta}(\hat\theta_{T(m),2} > \tilde\theta_2 - M, n_{T(m),2} < N).$$

Note that

$$\mathbb{P}^\pi_{\tilde\theta}(\hat\theta_{T(m),2} > \tilde\theta_2 - M, n_{T(m),2} < N)$$
$$= p - \sum_{n=1}^{N-1}\mathbb{P}^\pi_{\tilde\theta}(\hat\theta_{T(m),2} \leq \tilde\theta_2 - M, n_{T(m),2} = n)$$
$$\geq p - \sum_{n=1}^{N-1}\exp(-\frac{nM^2}{2\sigma^2}) \geq p - N\exp(-\frac{M^2}{2\sigma^2}) = q > 0.$$

Therefore, there exists a constant positive probability such that $\pi$ pulls arm 2 no more than $N$ times under $\theta$. As a result, $\pi$ incurs a linear expected regret under $\theta$, leading to a contradiction.

$\square$

**Proof of Theorem 3.1.**

1. Fix $\delta > \alpha$ and $\gamma > \alpha + \delta - 1$. We consider the environment where the noise $\epsilon$ is gaussian with standard deviation $\sigma$. Let $\theta_1 = 1/2$. Let $\theta(T) = (\theta_1, \theta_2(T))$ and $\tilde\theta(T) = (\theta_1, \tilde\theta_2(T))$, where $\theta_2(T) = \theta_1 + \frac{1}{2T^{1-\delta}}$ and $\tilde\theta_2(T) = \theta_1 - \frac{1}{2T^{1-\delta}}$. Let

$$\xi \in (\alpha + 1 - \delta, 1 \wedge (\gamma + 2 - 2\delta)).$$

Such $\xi$ always exists because $\alpha < \delta$ and $\gamma + 2 - 2\delta > \alpha + \delta - 1 + 2 - 2\delta = \alpha + 1 - \delta$. For notation simplicity, we will write $\theta$ ($\tilde\theta$) instead of $\theta(T)$ ($\tilde\theta(T)$), but we must keep in mind that $\theta$ ($\tilde\theta$) is dependent on $T$. Define

$$E_T = \left\{|\hat\mu_{T,2} - \tilde\theta_2| \leq \varepsilon/n_2^\omega\right\}$$

where $\varepsilon > 0$ is a small number and $\omega = \frac{1-\delta}{\xi} \in [0, 1/2)$, and

$$F_T = \{n_2 \leq T^\xi\}.$$

Then under the environment $\tilde\theta$, we have

$$\mathbb{P}^\pi_{\tilde\theta}(\bar{F}_T) = \mathbb{P}^\pi_{\tilde\theta}(n_2 > T^\xi) \leq \frac{\mathbb{E}^\pi_{\tilde\theta}[n_2]}{T^\xi} \leq \frac{\mathbb{E}[R^\pi_{\tilde\theta}(T)]}{T^{\xi+\delta-1}} \leq \frac{\sup_{\theta'}\mathbb{E}[R^\pi_{\theta'}(T)]}{T^{\xi+\delta-1}} \longrightarrow 0$$

as $T \to +\infty$. Combined with Lemma A.1, we have

$$\liminf_T \mathbb{P}_{\tilde{\theta}}^\pi(E_T, F_T) = 1.$$

Let $c' \in (0, 1/2)$. Now

$$\mathbb{P}\left(R_\theta^\pi(T) \geq c'T^\delta\right)$$
$$\geq \mathbb{P}_\theta^\pi(n_1 \geq 2c'T)$$
$$\geq \mathbb{P}_\theta^\pi(n_2 \leq (1 - 2c')T)$$
$$\geq \mathbb{P}_\theta^\pi(n_2 \leq T^\xi)$$
$$\geq \mathbb{P}_\theta^\pi(E_T, F_T)$$
$$= \mathbb{E}_\theta^\pi[\mathbb{1}\{E_T F_T\}]$$
$$= \mathbb{E}_{\tilde{\theta}}^\pi\left[\exp\left(\sum_{n=1}^{n_2} \frac{(X_{t_2(n),2} - \tilde{\theta}_2)^2 - (X_{t_2(n),2} - \theta_2)^2}{2\sigma^2}\right)\mathbb{1}\{E_T F_T\}\right]$$
$$= \mathbb{E}_{\tilde{\theta}}^\pi\left[\exp\left(n_2\left(\frac{\tilde{\theta}_2^2 - \theta_2^2}{2\sigma^2} + \frac{(\theta_2 - \tilde{\theta}_2)\hat{\theta}_{T,2}}{\sigma^2}\right)\right)\mathbb{1}\{E_T F_T\}\right]$$
$$\geq \mathbb{E}_{\tilde{\theta}}^\pi\left[\exp\left(n_2\left(\frac{\tilde{\theta}_2^2 - \theta_2^2}{2\sigma^2} + \frac{(\theta_2 - \tilde{\theta}_2)(\tilde{\theta}_2 - \varepsilon/n_2^\omega)}{\sigma^2}\right)\right)\mathbb{1}\{E_T F_T\}\right]$$
$$= \mathbb{E}_{\tilde{\theta}}^\pi\left[\exp\left(-n_2\frac{(\tilde{\theta}_2 - \theta_2)^2}{2\sigma^2} - \frac{\varepsilon n_2^{1-\omega}(\theta_2 - \tilde{\theta}_2)}{\sigma^2}\right)\mathbb{1}\{E_T F_T\}\right]$$
$$\geq \mathbb{E}_{\tilde{\theta}}^\pi\left[\exp\left(-T^\xi\frac{(\tilde{\theta}_2 - \theta_2)^2}{2\sigma^2} - \frac{\varepsilon T^{\xi+\delta-1}(\theta_2 - \tilde{\theta}_2)}{\sigma^2}\right)\mathbb{1}\{E_T F_T\}\right]$$
$$= \exp(-T^{\xi+2\delta-2}/2\sigma^2 - \varepsilon T^{\xi+2\delta-2}/\sigma^2)\mathbb{P}_{\tilde{\theta}}^\pi(E_T, F_T).$$

Notice that $\xi + 2\delta - 2$ can be arbitrarily close to $\alpha + \delta - 1$ and $\varepsilon > 0$ can be arbitrary. Therefore, we have

$$\liminf_T \frac{\ln\left\{\sup_{\theta'} \mathbb{P}(R_{\theta'}^\pi(T) \geq cT^\delta)\right\}}{T^\gamma} \geq \liminf_T \frac{-T^{\xi+2\delta-2}/2\sigma^2}{T^\gamma} = 0.$$

Since $\ln\left\{\sup_\theta \mathbb{P}(R_\theta^\pi(T) \geq cT^\delta)\right\} \leq 0$ always holds, we obtain the result.

2. Fix $\delta > \beta$ and $\gamma > \beta$. We consider the environment where the noise $\epsilon$ is gaussian with standard deviation $\sigma$. Let $\theta_1 = 1/2$. Let $\theta = (\theta_1, \theta_2)$ and $\tilde{\theta} = (\theta_1, \tilde{\theta}_2)$, where $\theta_2 = \theta_1 + \frac{1}{2}$ and $\tilde{\theta}_2 = \theta_1 - \frac{1}{2}$. Let $c' \in (c, 1/2)$. Define

$$E_T = \left\{|\hat{\mu}_{T,2} - \tilde{\theta}_2| \leq \varepsilon\right\}$$

where $\varepsilon > 0$ is a small number, and

$$F_T = \{n_2 \leq T^\xi\}.$$

with $\xi \in (\beta, \gamma)$. Under the environment $\tilde{\theta}$, we have

$$\mathbb{P}_{\tilde{\theta}}^\pi(\bar{F}_T) = \mathbb{P}_{\tilde{\theta}}^\pi\left(n_2 > T^\xi\right) \leq \frac{\mathbb{E}_{\tilde{\theta}}^\pi[n_2]}{T^\xi} \leq \frac{2\mathbb{E}[R_{\tilde{\theta}}^\pi(T)]}{T^\xi}.$$

Combined with Lemma A.1, we have

$$\liminf_T \mathbb{P}_{\tilde{\theta}}^\pi(E_T, F_T) \geq 1 - \limsup_T \frac{2\mathbb{E}[R_{\tilde{\theta}}^\pi(T)]}{T^\xi} = 0. \tag{8}$$

Take $c' \in (c, 1/2)$. Now

$$\mathbb{P}\left(R_\theta^\pi(T) \geq cT^\delta\right)$$

$$\geq \mathbb{P}\left(R_\theta^\pi(T) \geq c'T\right)$$

$$\geq \mathbb{P}_\theta^\pi(n_1 \geq 2c'T)$$

$$\geq \mathbb{P}_\theta^\pi(n_2 \leq (1-2c')T)$$

$$\geq \mathbb{P}_\theta^\pi(n_2 \leq T^\xi)$$

$$\geq \mathbb{P}_\theta^\pi(E_T, F_T)$$

$$= \mathbb{E}_\theta^\pi[\mathbb{1}\{E_T F_T\}]$$

$$= \mathbb{E}_{\tilde{\theta}}^\pi\left[\exp\left(\sum_{n=1}^{n_2} \frac{(X_{t_2(n),2} - \tilde{\theta}_2)^2 - (X_{t_2(n),2} - \theta_2)^2}{2\sigma^2}\right)\mathbb{1}\{E_T F_T\}\right]$$

$$= \mathbb{E}_{\tilde{\theta}}^\pi\left[\exp\left(n_2\left(\frac{\tilde{\theta}_2^2 - \theta_2^2}{2\sigma^2} + \frac{(\theta_2 - \tilde{\theta}_2)\hat{\theta}_{T,2}}{\sigma^2}\right)\right)\mathbb{1}\{E_T F_T\}\right]$$

$$\geq \mathbb{E}_{\tilde{\theta}}^\pi\left[\exp\left(n_2\left(\frac{\tilde{\theta}_2^2 - \theta_2^2}{2\sigma^2} + \frac{(\theta_2 - \tilde{\theta}_2)(\tilde{\theta}_2 - \varepsilon)}{\sigma^2}\right)\right)\mathbb{1}\{E_T F_T\}\right]$$

$$= \mathbb{E}_{\tilde{\theta}}^\pi\left[\exp\left(n_2\left(-\frac{(\tilde{\theta}_2 - \theta_2)^2}{2\sigma^2} - \frac{\varepsilon(\theta_2 - \tilde{\theta}_2)}{\sigma^2}\right)\right)\mathbb{1}\{E_T F_T\}\right]$$

$$\geq \mathbb{E}_{\tilde{\theta}}^\pi\left[\exp\left(T^\xi\left(-\frac{(\tilde{\theta}_2 - \theta_2)^2}{2\sigma^2} - \frac{\varepsilon(\theta_2 - \tilde{\theta}_2)}{\sigma^2}\right)\right)\mathbb{1}\{E_T F_T\}\right]$$

$$= \exp(-T^\xi(1/2\sigma^2 + \varepsilon/\sigma^2))\mathbb{P}_{\tilde{\theta}}^\pi(E_T, F_T).$$

Therefore,

$$\liminf_T \frac{\ln\left\{\sup_{\theta'} \mathbb{P}\left(R_{\theta'}^\pi(T) \geq cT\right)\right\}}{T^\gamma}$$

$$\geq \liminf_T \frac{\ln\left\{\exp(-T^\xi(1/2\sigma^2 + \varepsilon/\sigma^2))\mathbb{P}_{\tilde{\theta}}^\pi(E_T, F_T)\right\}}{T^\gamma}$$

$$= 0. \tag{9}$$

$$\square$$

# B  Proofs for Section 3.2 & 4

Since Theorem 3.3 and Proposition 3.4 are special cases of Theorem 4.1 and Proposition 4.2, respectively, we will only illustrate the proofs for 4.1 and Proposition 4.2.

**Proof of Lemma 4.3.** We only need to prove one side. We have

$$\mathbb{P}\left(E_n(k, k') \geq x; \tau_n \leq T; k, k' \in \mathcal{A}_n\right)$$

$$= \mathbb{P}\left(\sum_{m=1}^n \sum_{t=\tau_{m-1}+1}^{\tau_m} b_t(\mathbb{1}\{a_t = k\} - \mathbb{1}\{a_t = k'\}) + \sum_{m=1}^n \epsilon_{t_k(m),k} - \sum_{m=1}^n \epsilon_{t_{k'}(m),k'} \geq nx;\right.$$

$$\left.\tau_n \leq T; k, k' \in \mathcal{A}_n\right)$$

$$\leq \mathbb{P}\left(\sum_{m=1}^n \sum_{t=\tau_{m-1}+1}^{\tau_m} b_t(\mathbb{1}\{a_t = k\} - \mathbb{1}\{a_t = k'\}) \geq \frac{Bn}{B+2\sigma}x; \tau_n \leq T; k, k' \in \mathcal{A}_n\right) +$$

$$\mathbb{P}\left(\sum_{m=1}^n \epsilon_{t_k(m),k} \geq \frac{\sigma n}{B+2\sigma}x; \tau_n \leq T; k, k' \in \mathcal{A}_n\right) +$$

$$\mathbb{P}\left(\sum_{m=1}^n \epsilon_{t_{k'}(m),k'} \leq -\frac{\sigma n}{B+2\sigma}x; \tau_n \leq T; k, k' \in \mathcal{A}_n\right)$$

$$\leq \mathbb{P}\left(\sum_{m=1}^{n} \mathbb{1}\{k,k' \in \mathcal{A}_{m-1}\} \cdot \sum_{t=\tau_{m-1}\wedge T+1}^{\tau_m \wedge T} b_t(\mathbb{1}\{a_t = k\} - \mathbb{1}\{a_t = k'\}) \geq \frac{Bn}{B+2\sigma}x\right) +$$

$$\mathbb{P}\left(\sum_{m=1}^{n} \epsilon_{t_k(m),k} \geq \frac{\sigma n}{B+2\sigma}x; \tau_n \leq T; k,k' \in \mathcal{A}_n\right) +$$

$$\mathbb{P}\left(\sum_{m=1}^{n} \epsilon_{t_{k'}(m),k'} \leq -\frac{\sigma n}{B+2\sigma}x; \tau_n \leq T; k,k' \in \mathcal{A}_n\right)$$

$$\leq \exp\left(-\frac{\left(\frac{nBx}{B+2\sigma}\right)^2}{2B^2n}\right) + 2\exp\left(-\frac{\left(\frac{\sigma nx}{B+2\sigma}\right)^2}{2\sigma^2 n}\right) = 3\exp\left(-\frac{nx^2}{2(B+2\sigma)^2}\right).$$

The last inequality holds from Azuma's inequality for martingales. In fact, if we regard

$$X_m = \mathbb{1}\{k,k' \in \mathcal{A}_{m-1}\} \cdot \sum_{t=\tau_{m-1}\wedge T+1}^{\tau_m \wedge T} b_t(\mathbb{1}\{a_t = k\} - \mathbb{1}\{a_t = k'\})$$

and $\mathcal{F}_m$ as all the information received up to phase $m$ (that is, up to time $\tau_m \wedge T$), then since the *uniformly* random permutation at phase $m$ is independent from all previous information, we have

$$\mathbb{E}[X_m|\mathcal{F}_{m-1}] = 0 \quad \text{and} \quad |X_m| \leq 1 \text{ a.s.}$$

**Proof of Theorem 3.3.** Without loss of generality, we assume $\theta_1 = \theta_*$. Let $\pi = $ SEwRP and $x \geq K$. Define

$$\mathcal{A}^* = \{k \neq 1 : n_k \leq 1 + T/K\}.$$

For any $k \neq 1$, we let $S_k$ be the event defined as

$$S_k = \{\text{Arm 1 is not eliminated before arm } k\}.$$

Then

$$\bar{S}_k = \{\text{Arm 1 is eliminated before arm } k\}.$$

We prove the bounds for two scenarios separately.

**1. Worst-case scenario.** We have

$$\mathbb{P}\left(R_\theta^\pi(T) \geq x\right)$$

$$= \mathbb{P}\left(\sum_{k\in\mathcal{A}^*} n_k\Delta_k + \sum_{k\notin\mathcal{A}^*} n_k\Delta_k \geq x\right)$$

$$\leq \mathbb{P}\left(\sum_{k\in\mathcal{A}^*} (n_k-1)\Delta_k + \sum_{k\notin\mathcal{A}^*} (n_k-1)\Delta_k \geq x - K\right)$$

$$\leq \mathbb{P}\left(\left(\bigcup_{k\in\mathcal{A}^*}\left\{(n_k-1)\Delta_k \geq \frac{x-K}{2K}\right\}\right)\cup\left(\bigcup_{k\notin\mathcal{A}^*}\left\{\Delta_k \geq \frac{x-K}{2T}\right\}\right)\right)$$

$$\leq \sum_{k\neq 1}\left(\mathbb{P}\left((n_k-1)\Delta_k \geq \frac{x-K}{2K}, k \in \mathcal{A}^*\right) + \mathbb{P}\left(\Delta_k \geq \frac{x-K}{2T}, k \notin \mathcal{A}^*\right)\right)$$

The reason that the second inequality holds is as follows. To prove it, we only need to show that the following cannot holds:

$$(n_k-1)\Delta_k < \frac{x-K}{2K}, \quad \forall k \in \mathcal{A}^*; \qquad \Delta_k < \frac{x-K}{2T}, \quad \forall k \notin \mathcal{A}^*.$$

If not, then we have

$$\sum_{k\neq 1}(n_k-1)\Delta_k$$

$$= \sum_{k \in \mathcal{A}^*} (n_k - 1)\Delta_k + \sum_{k \notin \mathcal{A}^*} (n_k - 1)\Delta_k$$

$$< \frac{(x - K)|\mathcal{A}^*|}{2K} + \frac{x - K}{2}$$

$$\leq \frac{x - K}{2} + \frac{x - K}{2}$$

$$= x - K.$$

Therefore,

$$\mathbb{P}\left(R_\theta^\pi(T) \geq x\right)$$

$$\leq \sum_{k \neq 1} \left( \mathbb{P}\left( (n_k - 1)\Delta_k \geq \frac{x - K}{2K}, \; k \in \mathcal{A}^* \right) + \mathbb{P}\left( \Delta_k \geq \frac{x - K}{2T}, \; k \notin \mathcal{A}^* \right) \right)$$

$$= \sum_{k \neq 1} \mathbb{P}\left( (n_k - 1)\Delta_k \geq \frac{x - K}{2K}, \; k \in \mathcal{A}^*, S_k \right) + \sum_{k \neq 1} \mathbb{P}\left( (n_k - 1)\Delta_k \geq \frac{x - K}{2K}, \; k \in \mathcal{A}^*, \bar{S}_k \right)$$

$$+ \sum_{k \neq 1} \mathbb{P}\left( \Delta_k \geq \frac{x - K}{2T}, \; k \notin \mathcal{A}^*, S_k \right) + \sum_{k \neq 1} \mathbb{P}\left( \Delta_k \geq \frac{x - K}{2T}, \; k \notin \mathcal{A}^*, \bar{S}_k \right)$$

Fix $k \neq 1$. Now for each $k$, we consider bounding the four terms separately.

(a) $k \in \mathcal{A}^*$. With a slight abuse of notation, we let $n_0 = \lceil \frac{x - K}{2K\Delta_k} \rceil \leq n_k - 1$. Also,

$$\Delta_k \geq \frac{x - K}{2K(n_k - 1)} \geq \frac{(x - K)K}{2KT} = \frac{x - K}{2T}.$$

Then after completing phase $n_0$, arm $k$ is not eliminated and $\tau_{n_0} \leq T$. The event that $S_k$ holds means arm 1 is not eliminated either, and so

$$\hat{\mu}_{t_1(n_0),1} - \text{rad}(n_0) \leq \hat{\mu}_{t_k(n_0),k} + \text{rad}(n_0)$$

holds. We have

$$\mathbb{P}\left( (n_k - 1)\Delta_k \geq \frac{x - K}{2K}, \; k \in \mathcal{A}^*, S_k \right)$$

$$\leq \mathbb{P}\left( \hat{\mu}_{t_1(n_0),1} - \text{rad}(n_0) \leq \hat{\mu}_{t_k(n_0),k} + \text{rad}(n_0); \tau_{n_0} \leq T; 1, k \in \mathcal{A}_{n_0} \right) \mathbb{1}\left\{ \Delta_k \geq \frac{x - K}{2T} \right\}$$

$$= \mathbb{P}\left( \mu_1 + E_{n_0}(1, k) - \text{rad}(n_0) \leq \mu_k + \text{rad}(n_0); \tau_{n_0} \leq T; 1, k \in \mathcal{A}_{n_0} \right) \mathbb{1}\left\{ \Delta_k \geq \frac{x - K}{2T} \right\}$$

$$= \mathbb{P}\left( E_{n_0}(k, 1) \geq \Delta_k - 2\text{rad}(n_0); \tau_{n_0} \leq T; 1, k \in \mathcal{A}_{n_0} \right) \mathbb{1}\left\{ \Delta_k \geq \frac{x - K}{2T} \right\}$$

$$\leq 3 \exp\left( -n_0 \left( \Delta_k - 2\text{rad}(n_0) \right)_+^2 / 2(B + 2\sigma)^2 \right) \mathbb{1}\left\{ \Delta_k \geq \frac{x - K}{2T} \right\}$$

$$\leq 3 \exp\left( -n_0 \left( \Delta_k - 2\frac{\eta_1 (T/K)^\alpha \sqrt{\ln T}}{n_0} \right)_+^2 / 2(B + 2\sigma)^2 \right) \mathbb{1}\left\{ \Delta_k \geq \frac{x - K}{2T} \right\}$$

$$= 3 \exp\left( -n_0 \Delta_k^2 \left( 1 - \frac{2\eta_1 (T/K)^\alpha \sqrt{\ln T}}{n_0 \Delta_k} \right)_+^2 / 2(B + 2\sigma)^2 \right) \mathbb{1}\left\{ \Delta_k \geq \frac{x - K}{2T} \right\}$$

$$\leq 3 \exp\left( -\frac{(x - K)_+^2}{4KT} \left( 1 - \frac{4\eta_1 K^{1-\alpha} T^\alpha \sqrt{\ln T}}{x - K} \right)_+^2 / 2(B + 2\sigma)^2 \right)$$

$$\leq 3 \exp\left( -\frac{\left( x - K - 4\eta_1 K^{1-\alpha} T^\alpha \sqrt{\ln T} \right)_+^2}{8(B + 2\sigma)^2 KT} \right). \tag{10}$$

Then we bound $\mathbb{P}(n_k\Delta_k \geq (x-K)/2K, k \in \mathcal{A}^*, \bar{S}_k)$. Suppose that after $n$ phases, arm 1 is eliminated by arm $k'$ ($k'$ is not necessarily $k$). By the definition of $\bar{S}_k$, arm $k$ is not eliminated. Therefore, we have

$$\hat{\mu}_{t_{k'}(n),k'} - \mathrm{rad}(n) \geq \hat{\mu}_{t_1(n),1} + \mathrm{rad}(n)$$
$$\text{and} \tag{11}$$
$$\hat{\mu}_{t_k(n),k} + \mathrm{rad}(n) \geq \hat{\mu}_{t_1(n),1} + \mathrm{rad}(n)$$

holds simultaneously. The first inequality holds because arm 1 is eliminated. The second inequality holds because arm $k$ is not eliminated. Now for fixed $n$,

$$\mathbb{P}\left(\exists k' \in \mathcal{A}_n : (11) \text{ happens}; \Delta_k \geq \frac{x-K}{2T}; \tau_n \leq T; 1, k \in \mathcal{A}_n\right)$$

$$\leq \mathbb{P}\left(\exists k' \in \mathcal{A}_n : \hat{\mu}_{t_{k'}(n),k'} - \mathrm{rad}(n) \geq \hat{\mu}_{t_1(n),1} + \mathrm{rad}(n); \tau_n \leq T; 1, k \in \mathcal{A}_n\right)$$

$$\wedge \mathbb{P}\left(\hat{\mu}_{t_k(n),k} + \mathrm{rad}(n) \geq \hat{\mu}_{t_1(n),1} + \mathrm{rad}(n); \Delta_k \geq \frac{x-K}{2T}; \tau_n \leq T; 1, k \in \mathcal{A}_n\right)$$

$$\leq \mathbb{P}\left(\exists k' \in \mathcal{A}_n : E_n(k',1) \geq \mathrm{rad}(n); \tau_n \leq T; 1, k \in \mathcal{A}_n\right)$$

$$\wedge \mathbb{P}\left(E_n(k,1) \geq \Delta_k; \Delta_k \geq \frac{x-K}{2T}; \tau_n \leq T; 1, k \in \mathcal{A}_n\right)$$

$$\leq \left(\sum_{k' \neq 1} \mathbb{P}\left(E_n(k',1) \geq \mathrm{rad}(n); \tau_n \leq T; 1, k' \in \mathcal{A}_n\right)\right) \wedge$$

$$\mathbb{P}\left(E_n(k,1) \geq \frac{x-K}{2T}; \tau_n \leq T; 1, k \in \mathcal{A}_n\right)$$

$$\leq \left(3K \exp\left(-\frac{\eta_1^2(T/K)^{2\alpha}\ln T}{2(B+2\sigma)^2 n}\right) \vee \right.$$

$$\left. 3K \exp\left(-\frac{\eta_2^2 T^\beta \ln T}{2(B+2\sigma)^2}\right)\right) \wedge 3K \exp\left(-\frac{n(x-K)_+^2}{8(B+2\sigma)^2 T^2}\right)$$

$$\leq 3K \exp\left(-\left(\frac{\eta_1^2(T/K)^{2\alpha}\ln T}{2(B+2\sigma)^2 n} \vee \frac{n(x-K)_+^2}{8(B+2\sigma)^2 T^2}\right)\right) \vee 3K \exp\left(-\frac{\eta_2^2 T^\beta \ln T}{2(B+2\sigma)^2}\right)$$

$$\leq 3K \exp\left(-\frac{(x-K)_+ \eta_1 \sqrt{\ln T}}{4(B+2\sigma)^2 K^\alpha T^{1-\alpha}}\right) \vee 3K \exp\left(-\frac{\eta_2^2 T^\beta \ln T}{2(B+2\sigma)^2}\right)$$

$$= 3K \exp\left(-\frac{\sqrt{\ln T}}{2(B+2\sigma)^2}\left(\frac{\eta_1(x-K)_+}{2K^\alpha T^{1-\alpha}} \wedge \eta_2^2 T^\beta \sqrt{\ln T}\right)\right).$$

Therefore,

$$\mathbb{P}((n_k-1)\Delta_k \geq (x-K)/2K, \bar{S}_k, k \in \mathcal{A}^*)$$
$$= \mathbb{P}\left(\exists n \leq T : \exists k' \in \mathcal{A}_n : (11) \text{ happens}; (n_k-1)\Delta_k \geq (x-K)/2K, k \in \mathcal{A}^*; \right.$$
$$\left. \tau_n \leq T; 1, k \in \mathcal{A}_n\right)$$

$$\leq \sum_{n=1}^{T} \mathbb{P}\left(\exists k' \in \mathcal{A}_n : (11) \text{ happens}; \Delta_k \geq \frac{x-K}{2T}; \tau_n \leq T; 1, k \in \mathcal{A}_n\right)$$

$$\leq 3KT \exp\left(-\frac{\sqrt{\ln T}}{2(B+2\sigma)^2}\left(\frac{\eta_1(x-K)_+}{2K^\alpha T^{1-\alpha}} \wedge \eta_2^2 T^\beta \sqrt{\ln T}\right)\right). \tag{12}$$

(b) $k \notin \mathcal{A}^*$. With a slight abuse of notation, we let $n_0 = \lceil \frac{T}{K} \rceil \leq n_k - 1$. Also,

$$\Delta_k \geq \frac{x-K}{2T}.$$

Then after completing phase $n_0$, arm $k$ is not eliminated and $\tau_{n_0} \leq T$. The event that $S_k$ holds means arm 1 is not eliminated either, and so

$$\hat{\mu}_{t_1(n_0),1} - \mathrm{rad}(n_0) \leq \hat{\mu}_{t_k(n_0),k} + \mathrm{rad}(n_0)$$

holds. We have

$$\mathbb{P}\left(\Delta_k \geq \frac{x-K}{2T}, \, k \notin \mathcal{A}^*, S_k\right)$$

$$\leq \mathbb{P}\left(\hat{\mu}_{t_1(n_0),1} - \mathrm{rad}(n_0) \leq \hat{\mu}_{t_k(n_0),k} + \mathrm{rad}(n_0); \tau_{n_0} \leq T; 1, k \in \mathcal{A}_{n_0}\right)\mathbb{1}\left\{\Delta_k \geq \frac{x-K}{2T}\right\}$$

$$= \mathbb{P}\left(\mu_1 + E_{n_0}(1,k) - \mathrm{rad}(n_0) \leq \mu_k + \mathrm{rad}(n_0); \tau_{n_0} \leq T; 1, k \in \mathcal{A}_{n_0}\right)\mathbb{1}\left\{\Delta_k \geq \frac{x-K}{2T}\right\}$$

$$= \mathbb{P}\left(E_{n_0}(k,1) \geq \Delta_k - 2\mathrm{rad}(n_0); \tau_{n_0} \leq T; 1, k \in \mathcal{A}_{n_0}\right)\mathbb{1}\left\{\Delta_k \geq \frac{x-K}{2T}\right\}$$

$$\leq 3\exp\left(-n_0\left(\Delta_k - 2\mathrm{rad}(n_0)\right)_+^2 / 2(B+2\sigma)^2\right)\mathbb{1}\left\{\Delta_k \geq \frac{x-K}{2T}\right\}$$

$$\leq 3\exp\left(-n_0\left(\Delta_k - 2\frac{\eta_1(T/K)^\alpha\sqrt{\ln T}}{n_0}\right)_+^2 \middle/ 2(B+2\sigma)^2\right)\mathbb{1}\left\{\Delta_k \geq \frac{x-K}{2T}\right\}$$

$$= 3\exp\left(-\frac{T}{K}\left(\frac{x-K}{2T} - \frac{2\eta_1(T/K)^\alpha\sqrt{\ln T}}{T/K}\right)_+^2 \middle/ 2(B+2\sigma)^2\right)$$

$$\leq 3\exp\left(\frac{\left(x-K-4\eta_1 K^{1-\alpha}T^\alpha\sqrt{\ln T}\right)_+^2}{8(B+2\sigma)^2 KT}\right). \tag{13}$$

Then we bound $\mathbb{P}(\Delta_k \geq (x-K)/2T, k \notin \mathcal{A}^*, \bar{S}_k)$. The procedure is nearly the same as in the case where $k \in \mathcal{A}^*$. Suppose that after $n$ phases, arm 1 is eliminated by arm $k'$ ($k'$ is not necessarily $k$). By the definition of $\bar{S}_k$, arm $k$ is not eliminated. Therefore, we have

$$\hat{\mu}_{t_{k'}(n),k'} - \mathrm{rad}(n) \geq \hat{\mu}_{t_1(n),1} + \mathrm{rad}(n) \quad \text{and} \quad \hat{\mu}_{t_k(n),k} + \mathrm{rad}(n) \geq \hat{\mu}_{t_1(n),1} + \mathrm{rad}(n) \tag{14}$$

holds simultaneously. The first inequality holds because arm 1 is eliminated. The second inequality holds because arm $k$ is not eliminated. Now for fixed $n$,

$$\mathbb{P}\left(\exists k' \in \mathcal{A}_n : (14) \text{ happens}; \Delta_k \geq \frac{x-K}{2T}; \tau_n \leq T; 1, k \in \mathcal{A}_n\right)$$

$$\leq \mathbb{P}\left(\exists k' \in \mathcal{A}_n : \hat{\mu}_{t_{k'}(n),k'} - \mathrm{rad}(n) \geq \hat{\mu}_{t_1(n),1} + \mathrm{rad}(n); \tau_n \leq T; 1, k \in \mathcal{A}_n\right)$$

$$\wedge \mathbb{P}\left(\hat{\mu}_{t_k(n),k} + \mathrm{rad}(n) \geq \hat{\mu}_{t_1(n),1} + \mathrm{rad}(n); \Delta_k \geq \frac{x-K}{2T}; \tau_n \leq T; 1, k \in \mathcal{A}_n\right)$$

$$\leq \mathbb{P}\left(\exists k' \in \mathcal{A}_n : E_n(k',1) \geq \mathrm{rad}(n); \tau_n \leq T; 1, k \in \mathcal{A}_n\right)$$

$$\wedge \mathbb{P}\left(E_n(k,1) \geq \Delta_k; \Delta_k \geq \frac{x-K}{2T}; \tau_n \leq T; 1, k \in \mathcal{A}_n\right)$$

$$\leq \left(\sum_{k' \neq 1}\mathbb{P}\left(E_n(k',1) \geq \mathrm{rad}(n); \tau_n \leq T; 1, k' \in \mathcal{A}_n\right)\right) \wedge$$

$$\mathbb{P}\left(E_n(k,1) \geq \frac{x-K}{2T}; \tau_n \leq T; 1, k \in \mathcal{A}_n\right)$$

$$\leq \left(3K\exp\left(-\frac{\eta_1^2(T/K)^{2\alpha}\ln T}{2(B+2\sigma)^2 n}\right) \vee 3K\exp\left(-\frac{\eta_2^2 T^\beta \ln T}{2(B+2\sigma)^2}\right)\right) \wedge$$

$$3K\exp\left(-\frac{n(x-K)_+^2}{8(B+2\sigma)^2 T^2}\right)$$

$$\leq 3K\exp\left(-\left(\frac{\eta_1^2(T/K)^{2\alpha}\ln T}{2(B+2\sigma)^2 n} \vee \frac{n(x-K)_+^2}{8(B+2\sigma)^2 T^2}\right)\right) \vee 3K\exp\left(-\frac{\eta_2^2 T^\beta \ln T}{2(B+2\sigma)^2}\right)$$

$$\leq 3K \exp\left(-\frac{(x-K)_+ \eta_1 \sqrt{\ln T}}{4(B+2\sigma)^2 K^\alpha T^{1-\alpha}}\right) \vee 3K \exp\left(-\frac{\eta_2^2 T^\beta \ln T}{2(B+2\sigma)^2}\right)$$

$$= 3K \exp\left(-\frac{\sqrt{\ln T}}{2(B+2\sigma)^2}\left(\frac{\eta_1(x-K)_+}{2K^\alpha T^{1-\alpha}} \wedge \eta_2^2 T^\beta \sqrt{\ln T}\right)\right).$$

Therefore,

$$\mathbb{P}((n_k-1)\Delta_k \geq (n_k-1)(x-K)/2T, \bar{S}_k, k \notin \mathcal{A}^*)$$
$$= \mathbb{P}\left(\exists n \leq T : \exists k' \in \mathcal{A}_n : (14) \text{ happens}; (n_k-1)\Delta_k \geq (x-K)/2K, k \in \mathcal{A}^*;\right.$$
$$\left.\tau_n \leq T; 1, k \in \mathcal{A}_n\right)$$
$$\leq \sum_{n=1}^{T} \mathbb{P}\left(\exists k' \in \mathcal{A}_n : (14) \text{ happens}; \Delta_k \geq \frac{x-K}{2T}; \tau_n \leq T; 1, k \in \mathcal{A}_n\right)$$
$$\leq 3KT \exp\left(-\frac{\sqrt{\ln T}}{2(B+2\sigma)^2}\left(\frac{\eta_1(x-K)_+}{2K^\alpha T^{1-\alpha}} \wedge \eta_2^2 T^\beta \sqrt{\ln T}\right)\right). \tag{15}$$

Note that the equations above hold for any instance $\theta$. Combining (10), (12), (13), (15) yields

$$\sup_\theta \mathbb{P}(R_\theta^\pi(T) \geq x) \leq 6K \exp\left(\frac{\left(x-K-4\eta_1 K^{1-\alpha}T^\alpha\sqrt{\ln T}\right)_+^2}{8(B+2\sigma)^2 KT}\right) +$$

$$6K^2 T \exp\left(-\frac{\sqrt{\ln T}}{2(B+2\sigma)^2}\left(\frac{\eta_1(x-K)_+}{2K^\alpha T^{1-\alpha}} \wedge \eta_2^2 T^\beta \sqrt{\ln T}\right)\right).$$

**2. Instance-dependent scenario.** We have

$$\mathbb{P}\left(R_\theta^\pi(T) \geq x\right)$$

$$= \mathbb{P}\left(\sum_{k:\Delta_k>0} n_k \Delta_k \geq x\right)$$

$$\leq \mathbb{P}\left(\sum_{k:\Delta_k>0} (n_k-1)\Delta_k \geq x-K\right)$$

$$\leq \mathbb{P}\left(\bigcup_{k:\Delta_k>0}\left\{(n_k-1)\Delta_k \geq \frac{(x-K)/\Delta_k}{\sum_{k':\Delta_{k'}>0} 1/\Delta_{k'}}\right\}\right)$$

$$\leq \sum_{k:\Delta_k>0} \mathbb{P}\left((n_k-1)\Delta_k \geq \frac{(x-K)/\Delta_k}{\sum_{k':\Delta_{k'}>0} 1/\Delta_{k'}}\right)$$

$$\leq \sum_{k:\Delta_k>0} \mathbb{P}\left((n_k-1)\Delta_k \geq \frac{(x-K)/\Delta_k}{\sum_{k':\Delta_{k'}>0} 1/\Delta_{k'}}, S_k\right) +$$

$$\sum_{k:\Delta_k>0} \mathbb{P}\left((n_k-1)\Delta_k \geq \frac{(x-K)/\Delta_k}{\sum_{k':\Delta_{k'}>0} 1/\Delta_{k'}}, \bar{S}_k\right)$$

Denote

$$\Delta_0 = \frac{1}{\sum_{k':\Delta_{k'}>0} 1/\Delta_{k'}}.$$

Fix $k : \Delta_k > 0$. Now for each $k$, with a slight abuse of notation, we let

$$n_0 = \left\lceil (x-K)/\Delta_k^2 \cdot \Delta_0 \right\rceil \leq n_k - 1$$

Then after completing phase $n_0$, arm $k$ is not eliminated and $\tau_{n_0} \leq T$. The event that $S_k$ holds means arm 1 is not eliminated either, and so

$$\hat{\mu}_{t_1(n_0),1} - \text{rad}(n_0) \leq \hat{\mu}_{t_k(n_0),k} + \text{rad}(n_0)$$

holds. We have

$$\mathbb{P}\left((n_k - 1)\Delta_k \geq \frac{(x - K)/\Delta_k}{\sum_{k':\Delta_{k'}>0} 1/\Delta_{k'}}, S_k\right)$$

$$\leq \mathbb{P}\left(\hat{\mu}_{t_1(n_0),1} - \text{rad}(n_0) \leq \hat{\mu}_{t_k(n_0),k} + \text{rad}(n_0); \tau_{n_0} \leq T; 1, k \in \mathcal{A}_{n_0}\right)$$

$$= \mathbb{P}\left(\mu_1 + E_{n_0}(1,k) - \text{rad}(n_0) \leq \mu_k + \text{rad}(n_0); \tau_{n_0} \leq T; 1, k \in \mathcal{A}_{n_0}\right)$$

$$= \mathbb{P}\left(E_{n_0}(k,1) \geq \Delta_k - 2\text{rad}(n_0); \tau_{n_0} \leq T; 1, k \in \mathcal{A}_{n_0}\right)$$

$$\leq 3\exp\left(-n_0\left(\Delta_k - 2\text{rad}(n_0)\right)_+^2 / 2(B + 2\sigma)^2\right)$$

$$\leq 3\exp\left(-n_0\left(\Delta_k - 2\frac{\eta_2\sqrt{T^\beta \ln T}}{\sqrt{n_0}}\right)_+^2 / 2(B + 2\sigma)^2\right)$$

$$= 3\exp\left(-\left(\sqrt{n_0\Delta_k}\sqrt{\Delta_k} - 2\eta_2\sqrt{T^\beta \ln T}\right)_+^2 / 2(B + 2\sigma)^2\right)$$

$$\leq 3\exp\left(-\left(\sqrt{(x-K)\Delta_0} - 2\eta_2\sqrt{T^\beta \ln T}\right)_+^2 / 2(B + 2\sigma)^2\right)$$

$$\leq 3\exp\left(-\frac{\left((x-K)\Delta_0 - 4\eta_2^2 T^\beta \ln T\right)_+}{2(B + 2\sigma)^2}\right). \tag{16}$$

Then we bound $\mathbb{P}\left((n_k - 1)\Delta_k \geq (x - K)\Delta_0/\Delta_k, \bar{S}_k\right)$. Suppose that after $n$ phases, arm 1 is eliminated by arm $k'$ ($k'$ is not necessarily $k$). By the definition of $\bar{S}_k$, arm $k$ is not eliminated. Therefore, we have

$$\hat{\mu}_{t_{k'}(n),k'} - \text{rad}(n) \geq \hat{\mu}_{t_1(n),1} + \text{rad}(n) \quad \text{and} \quad \hat{\mu}_{t_k(n),k} + \text{rad}(n) \geq \hat{\mu}_{t_1(n),1} + \text{rad}(n) \tag{17}$$

holds simultaneously. The first inequality holds because arm 1 is eliminated. The second inequality holds because arm $k$ is not eliminated. Now for fixed $n$,

$$\mathbb{P}\left(\exists k' \in \mathcal{A}_n : (17) \text{ happens}; (n_k - 1)\Delta_k \geq (x - K)\Delta_0/\Delta_k; \tau_n \leq T; 1, k \in \mathcal{A}_n\right)$$

$$\leq \mathbb{P}\left(\exists k' \in \mathcal{A}_n : \hat{\mu}_{t_{k'}(n),k'} - \text{rad}(n) \geq \hat{\mu}_{t_1(n),1} + \text{rad}(n); \tau_n \leq T; 1, k \in \mathcal{A}_n\right)$$

$$\qquad \wedge \mathbb{P}\left(\hat{\mu}_{t_k(n),k} + \text{rad}(n) \geq \hat{\mu}_{t_1(n),1} + \text{rad}(n); \tau_n \leq T; 1, k \in \mathcal{A}_n\right)$$

$$\leq \mathbb{P}\left(\exists k' \in \mathcal{A}_n : E_n(k',1) \geq \text{rad}(n); \tau_n \leq T; 1, k \in \mathcal{A}_n\right)$$

$$\qquad \wedge \mathbb{P}\left(E_n(k,1) \geq \Delta_k; \tau_n \leq T; 1, k \in \mathcal{A}_n\right)$$

$$\leq \left(\sum_{k':\Delta_{k'}>0} \mathbb{P}\left(E_n(k',1) \geq \text{rad}(n); \tau_n \leq T; 1, k' \in \mathcal{A}_n\right)\right) \wedge$$

$$\qquad \mathbb{P}\left(E_n(k,1) \geq \Delta_k; \tau_n \leq T; 1, k \in \mathcal{A}_n\right)$$

$$\leq \left(3K\exp\left(-\frac{\eta_1^2(T/K)^{2\alpha}\ln T}{2(B+2\sigma)^2 n}\right) \vee 3K\exp\left(-\frac{\eta_2^2 T^\beta \ln T}{2(B+2\sigma)^2}\right)\right) \wedge$$

$$\qquad 3K\exp\left(-\frac{n\Delta_k^2}{2(B+2\sigma)^2}\right)$$

$$\leq 3K\exp\left(-\left(\frac{\eta_1^2(T/K)^{2\alpha}\ln T}{2(B+2\sigma)^2 n} \vee \frac{n\Delta_k^2}{2(B+2\sigma)^2}\right)\right) \vee 3K\exp\left(-\frac{\eta_2^2 T^\beta \ln T}{2(B+2\sigma)^2}\right)$$

$$\leq 3K\exp\left(-\frac{\eta_1(T/K)^\alpha\sqrt{\ln T}\Delta_k}{2(B+2\sigma)^2}\right) \vee 3K\exp\left(-\frac{\eta_2^2 T^\beta \ln T}{2(B+2\sigma)^2}\right)$$

$$= 3K\exp\left(-\frac{\sqrt{\ln T}}{2(B+2\sigma)^2}\left(\eta_1(T/K)^\alpha\Delta_k \wedge \eta_2^2 T^\beta\sqrt{\ln T}\right)\right).$$

Therefore,

$$\mathbb{P}((n_k - 1)\Delta_k \geq (x - K)\Delta_0/\Delta_k, \bar{S}_k)$$

$$= \mathbb{P}\left(\exists n \leq T : \exists k' \in \mathcal{A}_n : (17) \text{ happens}; (n_k - 1)\Delta_k \geq (x - K)\Delta_0/\Delta_k, k \in \mathcal{A}^*; \right.$$
$$\left. \tau_n \leq T; 1, k \in \mathcal{A}_n \right)$$

$$\leq \sum_{n=1}^{T} \mathbb{P}\left(\exists k' \in \mathcal{A}_n : (17) \text{ happens}; (n_k - 1)\Delta_k \geq (x - K)\Delta_0/\Delta_k; \tau_n \leq T; 1, k \in \mathcal{A}_n \right)$$

$$\leq 3KT \exp\left(-\frac{\sqrt{\ln T}}{2(B + 2\sigma)^2} \left(\eta_1 (T/K)^\alpha \Delta_k \wedge \eta_2^2 T^\beta \sqrt{\ln T}\right)\right). \tag{18}$$

Combining (16) and (18) yields

$$\mathbb{P}(R_\theta^\pi(T) \geq x)$$

$$\leq 3K \exp\left(-\frac{((x - K)\Delta_0 - 4\eta_2^2 T^\beta \ln T)_+}{2(B + 2\sigma)^2}\right) +$$

$$3KT \sum_{k:\Delta_k > 0} \exp\left(-\frac{\sqrt{\ln T}}{2(B + 2\sigma)^2} \left(\eta_1 (T/K)^\alpha \Delta_k \wedge \eta_2^2 T^\beta \sqrt{\ln T}\right)\right).$$

$\square$

**Proof of Proposition 3.4.** We have

$$\mathbb{E}[R_\theta^\pi(T)] \leq x + T \cdot \mathbb{P}(R_\theta^\pi(T) \geq x).$$

1. For worst-case scenario, let $x = K + C(\eta_1 \vee B^2/\eta_1 \vee \sigma^2/\eta_1)K^{1-\alpha}T^\alpha\sqrt{\ln T}$ with the constant $C$ being moderately large, then we have

$$T \cdot \mathbb{P}(R_\theta^\pi(T) \geq x)$$

$$\leq 6KT \exp\left(-\Omega\left(\frac{K^{2-2\alpha}T^{2\alpha}\ln T}{KT}\right)\right)$$

$$+ 6K^2T^2 \exp\left(-\Omega\left(\frac{K^{1-\alpha}T^\alpha \ln T}{K^\alpha T^{1-\alpha}}\right)\right) + 6K^2T^2 \exp\left(-\Omega\left(\frac{\eta_2^2 T^\beta \ln T}{(B + 2\sigma)^2}\right)\right)$$

$$= O(1).$$

When $\beta = 0$, $\eta_2$ need to be $\Omega(1 \vee \sigma)$. Therefore,

$$\mathbb{E}[R_\theta^\pi(T)] = O\left((\eta_1 \vee B^2/\eta_1 \vee \sigma^2/\eta_1)K^{1-\alpha}T^\alpha\sqrt{\ln T}\right) + O(1)$$

$$= O\left((\eta_1 \vee B^2/\eta_1 \vee \sigma^2/\eta_1)K^{1-\alpha}T^\alpha\sqrt{\ln T}\right)$$

2. For instance-dependent scenario, let $x = K + C\eta_2^2(B^2 \vee \sigma^2)T^\beta \ln T/\Delta_0$ with the constant $C$ being moderately large, then we have

$$T \cdot \mathbb{P}(R_\theta^\pi(T) \geq x)$$

$$\leq 3KT \exp\left(-\Omega(\ln T)\right) +$$

$$3KT^2 \sum_{k:\Delta_k > 0} \exp\left(-\frac{\eta_1 (T/K)^\alpha \Delta_k}{2(B + 2\sigma)^2}\right) + 3K^2T^2 \exp\left(-\frac{\eta_2^2 T^\beta \ln T}{2(B + 2\sigma)^2}\right)$$

$$= O(1).$$

When $\beta = 0$, $\eta_2$ need to be $\Omega(B \vee \sigma)$. Note that the second term can be bounded by a constant only unrelated with $T$ (but related with $\{\Delta_k\}$) because

$$\sup_T \text{poly}(T) \cdot \exp(-\text{poly}(T)) = O(1).$$

Therefore,

$$\mathbb{E}\left[R_\theta^\pi(T)\right] = O\left(\eta_2^2(B \vee \sigma^2)T^\beta \ln T \sum_{k:\Delta_k>0} \frac{1}{\Delta_k}\right),$$

ignoring the constant terms.

$\square$

## C  Numerical Experiments

In this section, we conduct a brief account of numerical experiments. The purpose of these experiments is not to compare our proposed policy to standard MAB policies, because standard MAB policies do not balance tail risks and therefore have inferior performances regarding tail probabilities of incurring a large regret or low cumulative reward (see, e.g., [24]). Instead, the purpose of experiments is to better illustrate our theoretical findings and understand how tuning parameters affect the performance of our proposed policy.

We first consider a two armed bandit problem with $\theta = (0.2, 0.8), \sigma = 1, T = 500$ and Gaussian noise. The non-stationary baseline is set as $b_t = \sin t$. We test our proposed policy SEwRP with difference choices of parameters $(\alpha, \beta, \eta_1, \eta_2)$ in the bonus term (2). We let $\alpha \in \{1/2, 2/3\}$, $\beta \in \{0, 1/6, 1/3, 1/2\}$. The tuning parameter $\eta_i$ $(i = 1, 2)$ has 4 choices: $\eta \in \{0.1, 0.2, 0.4, 0.8\}$. That said, for a fixed $\alpha$, we have $(4 \times 4 \times 4) = 64$ different versions of the policy SEwRP. For each version of policy, we run $5000$ simulation paths and for each path we collect the cumulative reward. We plot the empirical distribution (histogram) for a policy's cumulative reward in Figure 1. We also consider a $4$-armed bandit problem with $\theta = (0.2, 0.4, 0.6, 0.8), \sigma = 1, T = 500$ and Gaussian noise. Same as in the two-armed case, $\alpha \in \{1/2, 2/3\}, \beta \in \{0, 1/6, 1/3, 1/2\}$. The tuning parameter $\eta_i$ $(i = 1, 2)$ has 4 choices: $\eta \in \{0.1, 0.2, 0.4, 0.8\}$. For each tuple $(\alpha, \beta, \eta_1, \eta_2)$, we run $5000$ simulation paths and for each path we collect the cumulative reward. We plot the empirical distribution (histogram) for cumulative reward in Figure 2. We note that in both Figure 1 and 2, the first column mimics the standard SE policy. This is because we set $\beta = 0$ which makes rad$(n)$ dominated by its second term ($\propto \sqrt{\ln T/n}$).

**Discussion on $\eta_1$ and $\eta_2$.**

As shown in both figures, with $\alpha, \beta, \eta_2$ fixed, the reward distributions for different $\eta_1$ behave similarly regarding concentration. However, as $\eta_1$ increases, there is potentially a shift of the distribution mean. This is not surprising, since a large $\eta_1$ might make the policy too conservative. Also, when $\eta_2$ is small (say, $\eta_2 = 0.2$), the bonus term (2) is dominated by the second component, and thus the reward distributions behave nearly the same. With $\alpha, \beta, \eta_1$ fixed, as $\eta_2$ increases, the reward distribution becomes more concentrated. This can be explained by noticing that a larger $\eta_2$ gives more flexibility to the first component in the bonus term (2) which controls the worst-case regret tail behavior. However, if $\eta_1$ and $\beta$ are large (say, $\eta_1 = 0.8$ and $\beta = 1/3$), there is potentially a shift of the distribution mean. This is also not surprising, since a large $\eta_1$ indicates that (2) is dominated by its second component, while a large $\beta$ inflates the second component in (2).

**Discussion on $\alpha$ and $\beta$**

In both figures, With $\eta_1$ and $\eta_2$ fixed, as $\alpha$ increases, the reward distribution becomes more concentrated. This can be observed by noticing that when $\alpha = 1/2$, there is some part of distribution around 100 and it becomes even more significant for the 4-armed bandit case. This is consistent with our theoretical finding in Theorem 3.3, part 1, where we shows that tolerating more *sub-optimality* allows more light-tailed behaviour on extreme values. With $\eta_1$ and $\eta_2$ fixed, as $\beta$ increases, the reward distribution becomes more concentrated. That is, allowing more *inconsistency* gives rise to more concentration. This is in accordance with our theoretical finding in Theorem 3.3, part 2.

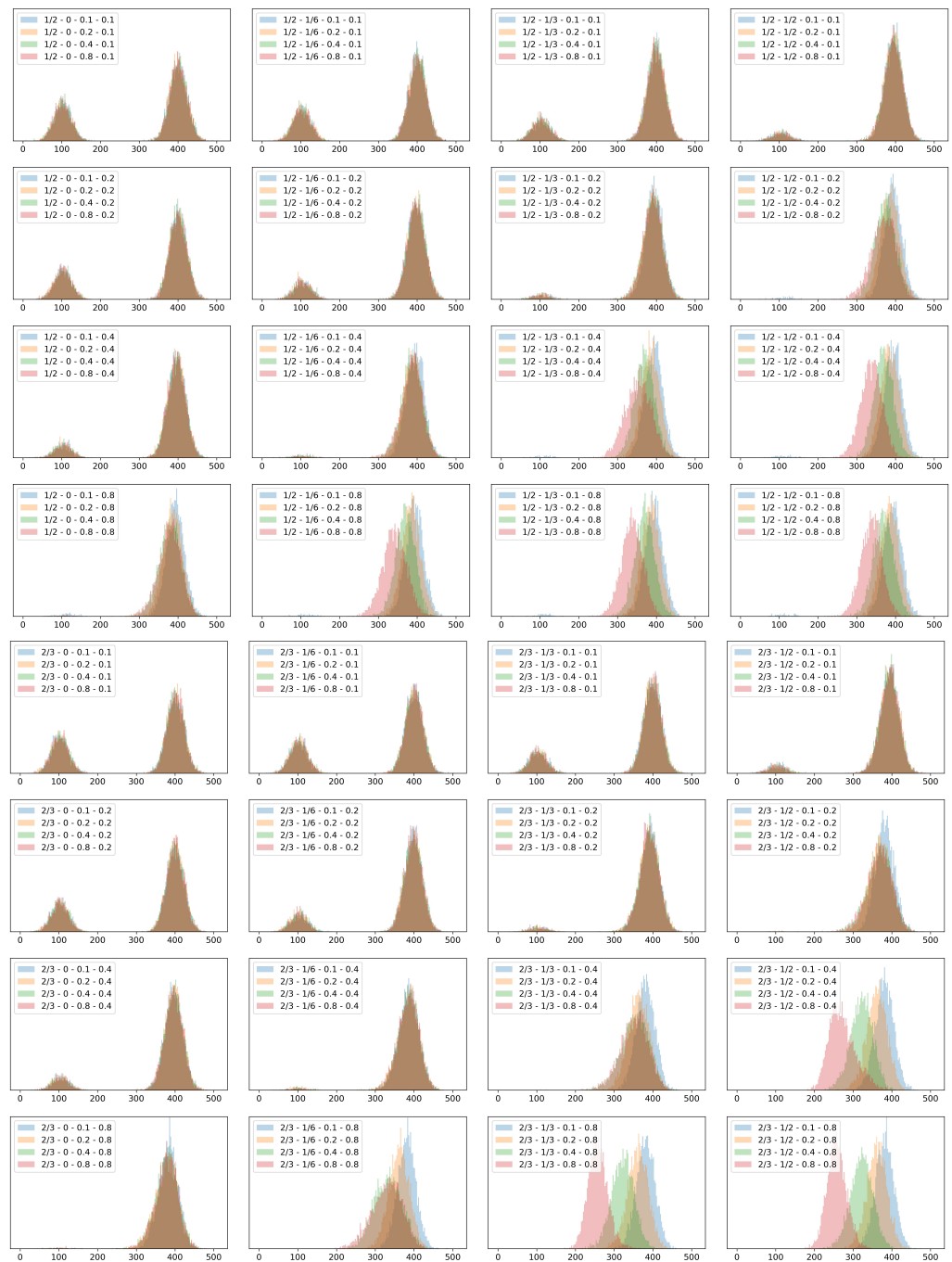

Figure 1: Empirical distribution for the cumulative reward (2-armed); labels displayed as $\alpha - \beta - \eta_1 - \eta_2$

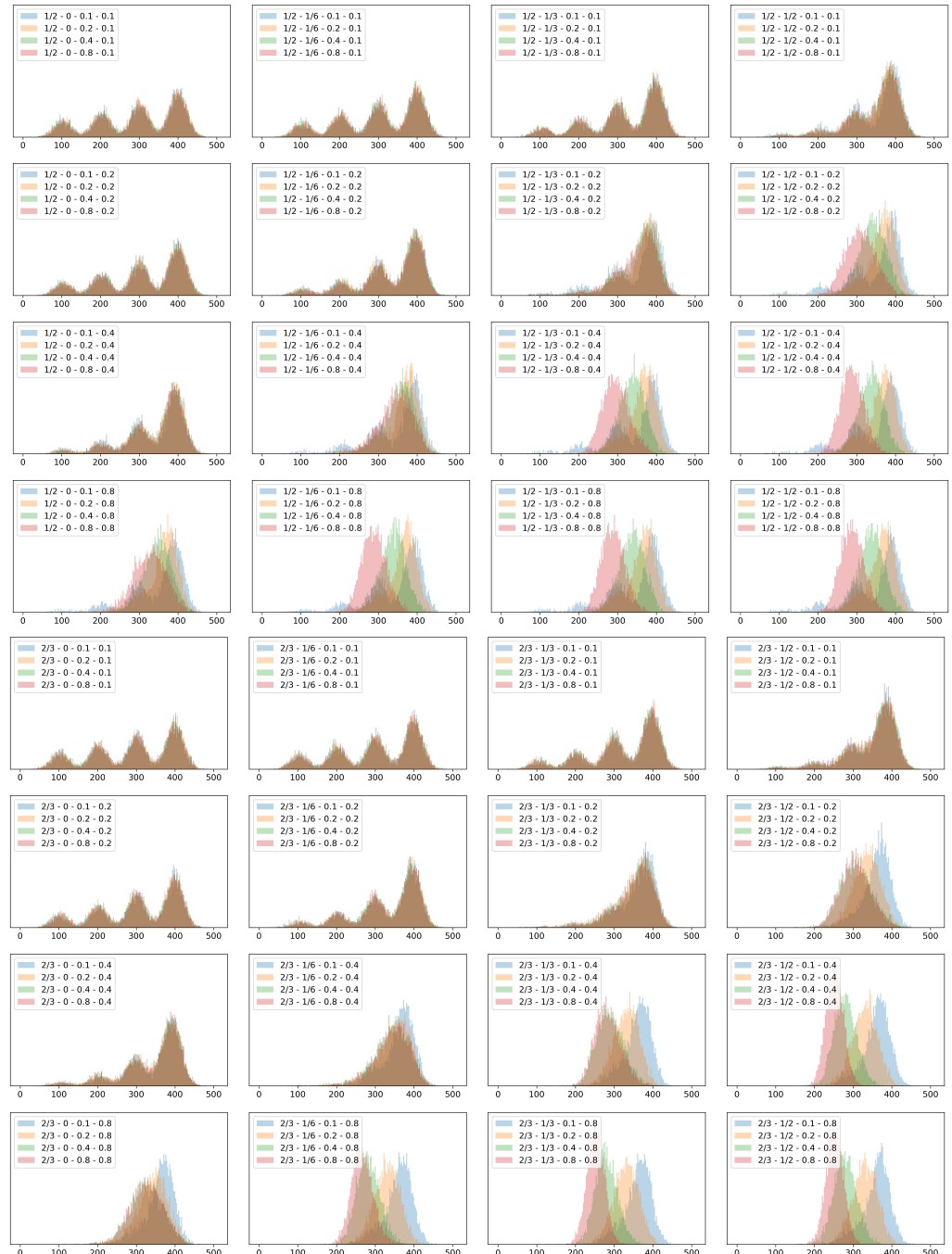

Figure 2: Empirical distribution for the cumulative reward (4-armed); labels displayed as $\alpha - \beta - \eta_1 - \eta_2$