# OpenReview forum: "Stochastic Multi-armed Bandits: Optimal Trade-off among Optimality, Consistency, and Tail Risk"
_NeurIPS.cc/2023/Conference — NeurIPS 2023 spotlight_

### Official Review · Reviewer_RNGE · 2023-06-25

**Soundness:** 3 good
**Presentation:** 2 fair
**Contribution:** 3 good
**Rating:** 7
**Confidence:** 4

**Summary:**

This paper tackles the problem of trading off problem-dependent and worst-case regret and "tail risk" of the regret in bandits. Here, the tail risk means the probability that the regret is larger than $\Omega(T^\delta)$ for some $\delta>0$. Recent studies have shown that the usual algorithms in bandits have unavoidably a linear regret with probability $O(T^{-1})$, and that without changing the scaling in $T$ of the confidence bounds of standard UCB algorithms one cannot hope to do better than a polynomially decreasing tail risk. Knowing if an exponentially decreasing tail risk is achievable is an interesting question in my opinion, and the authors answer favorably to it in the paper.

Providing a worst-case upper bound of $\Omega(T^\alpha)$ for $\alpha\geq1/2$ and a problem-dependent bound of $\Omega(T^\beta)$ for $\beta>0$, the authors determine the values $(\delta, \gamma)$ for which a bound of the kind $P(R(T) \geq cT^\delta)\leq \exp(-CT^\gamma)$ is achievable. They then provide a successive elimination strategy based on the UCB principle, that matches the best guarantees established given values of $(\alpha, \beta)$. They then provide an extension of their algorithm for a structured non-stationary setting.

**Strengths:**

The paper has several merits. In my opinion it tackles an interesting problem in an original way. Though it is a bit notation-heavy, overall every notion is clearly stated and relatively easy to understand. The theorems are also clear, both regarding the lower bounds and the upper bounds for the SEwRP algorithm. I think that the focus on this kind of successive elimination strategy is a good thing for the clarity of the paper.

Overall, I have a positive opinion of the paper, even if I believe that it requires some changes and clarifications before publication (see comments below).

**Weaknesses:**

Even if the paper is relatively easy to follow, the message of the paper is in my opinion not so clear and some changes may largely improve its clarity.

* In my opinion, starting from $(\alpha, \beta)$ to derive the bounds on the tail risk is not very natural. It seems more natural to first consider a given tail-risk constraint (e.g. imposed by a law-maker), and then try to propose an algorithm satisfying this constraint with the best regret guarantees. Furthermore, I tend to think that once we drop the possibility of logarithmic regret then the problem-dependent guarantees may not matter much. For this reason, I would tend to suggest a much simpler version of the paper where the authors would simply provide the best achievable worst case regret bound given a tail risk. In my opinion, this would make the paper just as interesting, and much clearer.

* From reading the main paper the intuition of the proofs and the technically difficult points are not very clear. I think that extended discussions on the results may bring value to the paper.

* Following previous point, to me it is not very clear that SEwRP is actually needed, and maybe a much simpler explore-then-commit strategy may satisfy all criteria.

* Section 4 does not bring much insights on the table and is more suited for the appendices in my opinion. I suggest to remove this section in order to provide more intuitions on the results and their proof in previous sections

* Practicality of the algorithms: it is folklore that the confidence bounds of UCB are already generally rather conservative. With such inflated confidence bounds, it is natural to wonder what happens when running the algorithm for reasonably large horizons. It is a bit disappointing that the experiment section is only in appendix, and only present regret distributions only (which is interesting) and not standard regret curves too. In particular, I wonder if the regret is really sub-linear for e.g $T=10^4$ and $9$ Bernoulli distributions with $\mu=0.5$ and one with $\mu=0.6$. Furthermore, a comparison of the performance of SewRP and standard bandit algorithms (or just standard UCB) is necessary in my opinion to also assess the "experimental" trade-off, even if their theoretical guarantees are not comparable and it is certain that the safe algorithm will perform worse on average.

The previous points are more opinions than objective statements, so I would be happy to discuss them with the authors in the discussion phase.

------------- Post-Rebuttal ----------------

The authors provided convincing answers to the previous points in their rebuttal, making me believe that none of these points is a major issue. Although I am still unsure on the practicality of the algorithm, I believe that the theoretical contribution of this work is strong enough so that it will be a nice Neurips paper.

**Questions:**

See previous section.

---

> ### Author Rebuttal · Authors · 2023-08-07
>
> We sincerely thank you for your valuable time and review. We like to provide responses to your comments, which are helpful and enlightening.
>
> - We’ve indeed tried adopting what you suggested, but ended up not doing so for two reasons.
>   - The relation between the tail probability and the tail threshold is complicated even for the worst-case scenario. Since the optimal log tail probability scales linearly with $x/T^{1-\alpha}$, if we were to specify the tail-risk constraint, it is unclear how we should set the threshold $x$ and how we should make the tail probability scale with $T$. Further, our tail bound holds for *any* threshold $x$, and thus it seems we should specify a group of tail-risk constraints instead of only one, which can possibly make the problem formulation a bit artificial.
>   - If instance-dependent consistency is considered, the tail probability becomes more complicated. We are not quite sure about the reviewer’s claim that “once we drop the possibility of logarithmic regret then the problem-dependent guarantees may not matter much”, but we would like to note that even if the instance-dependent scenario is not considered, consistency remains a significant effect on the tail probability (see Table 1). Further, considering consistency apart from optimality allows us to demonstrate how the adaptiveness of a policy to different instances affects the tail probabilities. This has not been studied in previous works.
>
> - We like to note the main proof ideas appeared in the paper (Line 237-269). The tail risk of an algorithm is incurred by two types of events: (i) spending too much time before correctly discarding a sub-optimal arm; (ii) wrongly discarding the optimal arm. For the first type of events, we focus on the phase $n_0$ when a sub-optimal arm is not eliminated by the optimal arm (see Line 248). For the second type of events, we focus on the phase $n_0$ when the optimal arm is eliminated by some other arm. In fact, it is particularly when bounding the probability of the second type of events that our new bonus design, compared to standard ones, allows light-tailed tail risk with optimal rate on the regret threshold $x$ and time horizon $T$. Finally, we note that the goal of defining $A^*$ in the worst-case scenario is to make sure our analysis aligns with our new bonus design (we have a $\sqrt{K}$ factor in the denominator) and our tail probabilities have optimal dependence on the number of arms $K$ (see also [24] in the paper and Line 242-243).
>
> - Applying the explore-then-commit (ETC) strategy is indeed a good idea to circumvent heavy-tailed risk. However, this approach has two main issues:
>    - From the regret expectation perspective, the ETC policy can only achieve $O(T^{2/3})$ worst-case expected regret bound. When $\alpha\in[1/2, 2/3)$, we still have to consider other types of policies. Further, it seems unlikely that without knowing the arm gaps $\Delta_k$, the explore-then-commit policy can achieve $\tilde O(T^{\alpha})$ worst-case regret and $\tilde O(T^{\beta})$ instance-dependent regret simultaneously with $\beta < \alpha$.
>    - From the regret tail risk perspective, the tail probability of incurring a large regret may not be optimal in the worst-case. Consider a simple $2$-armed bandit case with arm gap $\Delta$. An ETC policy with $m = \Theta(T^\alpha)$ ($\alpha\in[2/3, 1)$) steps of exploration has a probability of $\Omega(\exp(-m\Delta^2))$ committing to a wrong arm. In the worst-case scenario, let $\Delta = T^{(\alpha – 1)/2}$ and the regret threshold be $x = T^{(1+\alpha)/2} / 4 \in (m\Delta, (T-m)\Delta)$. Then if we incur a regret of $x$ it means we commit to a wrong arm after the exploration, which suggests $\mathbb P(R_\theta^\pi(T)>x) = \Omega(\exp(-m\Delta^2)) = \Omega(\exp(-T^{2\alpha-1})) = \omega(\exp(-x/T^{1-\alpha}))$.
>
>  - Section 4 is to show that our policy design reaches beyond the standard MAB case and is able to handle (structured) non-stationarities. We deem the results in Section 4 as an addition to Section 3 to demonstrate the generality of our results. In the next version, we are considering adding discussions about the proof of Section 3 and why ETC cannot apply in our setting, and briefly illustrate experiments.
>
> - On practicality of the algorithms:
>    - We note that in both Figure 1 and 2, the first column mimics the standard SE policy. This is because we set $\beta=0$ which makes $\text{rad}(n)$ dominated by its second term ($\propto\sqrt{\ln T/n}$). In Figure 1 and 2, one can observe that there is an experimental trade-off: Larger $\alpha$ (more sub-optimality) allows more light-tailed behavior on extreme values and larger $\beta$ (more inconsistency) gives rise to more concentration.
>    - Even with inflated bonus terms, the expected regret is comparable to standard policies. This phenomenon is indicated by comparing the blue distributions between the first and last column in Figure 1 and 2. We also run the experiment suggested by the reviewer. We take $\alpha\in\\{1/2, 2/3\\}$ and $\beta\in\\{0, 1/6, 1/3, 1/2\\}$. For each fixed $\alpha$ and $\beta$, we traverse $\eta_1$ and $\eta_2$ through $\\{0.05, 0.1, 0.2, 0.4, 0.8\\}$. For each $(\alpha, \beta, \eta_1, \eta_2)$, we run 1000 simulations and record the empirical mean regret, and so for each $(\alpha, \beta)$, we have 25 numbers. Then for each $(\alpha, \beta)$, we take the minimum of the 25 numbers (that is, we choose the best performed $(\eta_1, \eta_2)$). The results are listed below. We note again that when $\beta=0$, our policy can be regarded as the standard SE policy. As we can see from the results, for any fixed $\alpha$, as we increase $\beta$ and put more emphasis on the first term, the expected regret is approximately the same, suggesting that our policy does not sacrifice the expected regret.
> | $\alpha$ \ $\beta$| 0 | 1/6 | 1/3 | 1/2 |
> | --- | --- | --- | --- | --- |
> | 1/2 | 169.287 | 154.451 | 156.414 | 163.058 |
> | 2/3 | 271.661 | 276.071 | 272.956 | 271.200 |

---

> > ### Comment · Reviewer_RNGE · 2023-08-11
> > **Post-rebuttal comment**
> >
> > I acknowledge reading the other reviews and the rebuttal. Thank you very much for your insightful answers, that encouraged me to revise my score and vote for acceptance. More precisely,
> >
> > * Thank you for the precision on the presentation of the guarantees, I see your point and I now agree with you that your presentation may indeed be the better option.
> >
> > * Thank you for detailing the technical contribution.
> >
> > * Thank you for your detailed answer on ETC, which properly motivate using your method instead.
> >
> > *  I understand your point of view on section 4, my comment was a minor issue.
> >
> > To be completely honest I am still quite unconvinced on the practical aspect of the algorithms, and still believe in my initial intuition. Although I appreciate your effort in providing experimental result, I believe that a more extensive set of experiments may be necessary to convince me (and the problems that you consider in your short experiment section are rather easy). However, I do not believe that this is a major issue, and the theoretical contribution of this work is enough to make it a good Neurips paper (so please don't spend time on this).

---

> > > ### Author Response · Authors · 2023-08-12
> > >
> > > Thank you for your feedback and comments. We appreciate your comment on the practical aspect as well as the consideration. Indeed our experiments provide simple illustrations of algorithm performance, and we hope they may represent the performance for a wider range of scenarios. We look forward to exploring more in future work, as suggested by the reviewer, for a more extensive set of experiments. Thanks again!

---

### Official Review · Reviewer_URLi · 2023-07-02

**Soundness:** 3 good
**Presentation:** 3 good
**Contribution:** 3 good
**Rating:** 7
**Confidence:** 2

**Summary:**

This paper explores the stochastic multi-armed bandit (MAB) problem. The authors investigate the relationship between worst-case optimality, instance-dependent consistency, and light-tailed risk in policy design. Three main properties are considered for policy design: worst-case optimality, instance-dependent consistency, and light-tailed risk.

- The authors show the interplay between worst-case optimality, instance-dependent consistency, and light-tailed risk is characterized, indicating that relaxing the worst-case or instance-dependent regret order can lighten the regret tail in an information-theoretic way.

- A novel policy is designed that achieves an optimal trade-off among worst-case optimality, instance-dependent consistency, and light-tailed risk.

- The theory is generalized to a MAB model that allows for non-stationary baseline rewards, a common situation among all arms for each time period.

**Strengths:**

This paper makes contributions to the understanding of the stochastic multi-armed bandit problem and presents a novel policy design, which provides further insight into worst-case optimality, instance-dependent consistency, and light-tailed risk.

The authors have provided a detailed characterization of the interplays among worst-case optimality, instance-dependent consistency, and light-tailed risk. They successfully illustrate how different levels of these properties affect tail risk, and have determined an optimal trade-off among them.

The authors propose a unique policy Successive Elimination with Random Permutation (SEwRP) that achieves the optimal regret tail risk for any regret threshold, exhibiting desirable qualities in both worst-case and instance-dependent scenarios. This policy builds upon the concept of successive elimination, introducing novel bonus terms to balance the three key properties of policy design. For any given $\alpha$ and $\beta$, the proposed policy obtains optimal worst-case regret and instance-dependent regret, while also achieving the best possible regret tail probability for both scenarios.

The authors have successfully generalized their analysis to include a stochastic multi-armed bandit problem with non-stationary baseline rewards. This extension could prove useful in various applications dealing with structured non-stationarity.



**Weaknesses:**

Though the paper presents a novel and intriguing perspective, its primary reliance on theoretical analysis may be viewed as a limitation. Incorporating empirical validation of the proposed algorithm within the main body of the text could offer a more persuasive argument by substantiating the theoretical outcomes.

One limitation of this study is the unresolved question of whether the presented results can be extended to 'any-time' scenarios, where the policy has no prior knowledge of the time horizon $T$. The inability to validate the presented approach in the 'any-time' setting, a more realistic and complex scenario, leaves a gap in the research. This complexity should be addressed in future work to enhance the general applicability and robustness of the proposed algorithm.

**Questions:**

The paper is well-written, and the authors make the content accessible and easy to understand.

Given your paper's discussion in L228 on the phase transition regarding the size of the confidence interval in the design of the novel bonus term, could you elaborate on its interpretation and implications? Specifically, how does this phase transition impact the overall performance and robustness of the proposed policy? Additionally, could you provide further insight into the practical significance of this transition from the dominance of the second term to the first term in real-world applications?

**Limitations:**

Yes, the authors seem to have adequately addressed the limitations of the paper.

---

> ### Author Rebuttal · Authors · 2023-08-09
>
> We sincerely thank you for your valuable time and review. We would like to provide responses to your comments and questions, which we find very helpful to us.
>
> - Incorporating empirical validation: In the appendix, we provide detailed numerical experiments. We would like to emphasize that in both Figure 1 and 2, the first column indeed corresponds to the standard SE policy. This is because we set $\beta=0$ which makes $\text{rad}(n)$ dominated by the second term $\propto \sqrt{\ln T/n}$. One can observe that there is an experimental trade-off: Larger $\alpha$ (more sub-optimality) allows more light-tailed behavior on extreme values and larger $\beta$ (more inconsistency) gives rise to more concentration.
>
> - Insight of phase transition: The phase transition suggests that in the first phase where the second term dominates the first term, we focus more on exploration within the consistency constraint; in the second phase where the first term dominates the second term, we focus more exploitation within the optimality condition. While this is distinctive from the commit-then-explore paradigm where in the first phase we do pure exploration and in the second phase we do pure exploitation, our policy design suggests that in real-world practice, to achieve more light-tailed risk, it might be beneficial to have two different phases in the policy design: more exploration at the beginning, and more exploitation afterwards.

---

> > ### Comment · Reviewer_URLi · 2023-08-11
> > **Post-rebuttal comment**
> >
> > I appreciate the response from the authors. The response addressed all my questions and concerns, and I will keep my initial score for the paper.

---

> > > ### Author Response · Authors · 2023-08-12
> > >
> > > Thanks again for your time and comments!

---

### Official Review · Reviewer_2XJT · 2023-07-06

**Soundness:** 3 good
**Presentation:** 3 good
**Contribution:** 3 good
**Rating:** 7
**Confidence:** 4

**Summary:**

The submission studies the stochastic multi-armed bandits. The arm-selection policy is required to be worst-case optimal, instance-dependent consistency, and have low tail risks (worst-case and instance-dependent) simultaneously. Lower bounds for achieving these goals at the same time are provided in Theorem 3.1. The submission proposed a policy (SEwRP) that matches the lower bounds (Theorem 3.3 and Proposition 3.4). The main results are also generalized to non-stationary scenarios of baseline rewards (Theorem 4.1 and Proposition 4.2).

**Strengths:**

- (a) Studying the interplay between worst-case bound, consistency, and risk provides a new perspective to understanding MAB.
- (b) A novel confidence interval (rad(n)) is designed to address different criteria at different phases of the learning process.
- (c) Besides the rigorous analysis, the treatment for handling the confidence interval and the choice of tail event to tighten the bound constitute the technical contributions.
- (d) Fluent arguments and clarifying intuitions provide a comfortable reading experience.

**Weaknesses:**

- (e) The submission, with its supplementary, is a complete and self-content paper.

**Questions:**

- (f) In Algorithm 1, does i represent the number of elimination iterations, and t represent the number of arm pulls?
- (g) Is it correct that the reward's formulation (Line 124) contains the common Gaussian reward and normal reward?
- (h) What is the role of empirical regret? It is defined in Line 130 but does not appear in the key results and derivations.

**Limitations:**

Yes.

---

> ### Author Rebuttal · Authors · 2023-08-09
>
> We sincerely thank you for your valuable time and review. We would like to provide responses to the three questions (f) (g) and (h), which we find very helpful.
>
> - Yes, indeed you are correct. We will add more discussion for illustration in the next version.
>
> - Yes. Our model assumes subGaussian noises, and thus include the special case of pure Gaussian/normal rewards.
>
> - We define the empirical regret for completeness of describing the formulation. In practice, the DM can only observe the empirical reward $\sum_{t}r_{t, a_t}$, and thus the empirical regret can be naturally defined. We then focus on the pseudo regret by arguing that the sum of noise terms (genuine noise) is in general ignorable in the worst-case scenario or inevitable in the instance-dependent scenario (Line 141-145).

---

> > ### Comment · Reviewer_2XJT · 2023-08-14
> > **Post-rebuttal comment**
> >
> > I want to thank the authors' feedback on all reviews and all reviewers' comments. The feedback clarifies all my questions and provides insights from other reviews' comments. I would like to keep my original decision.

---

### Official Review · Reviewer_tbar · 2023-07-06

**Soundness:** 3 good
**Presentation:** 3 good
**Contribution:** 3 good
**Rating:** 7
**Confidence:** 4

**Summary:**

This paper presents an insightful investigation into the trade-off between optimality, measured by expected regret, and risk, defined as the probability of large regret, in the context of Multi-Armed Bandit problem algorithm design. The authors have made several significant contributions:

* They have demonstrated an "impossibility" result (Theorem 3.1), which indicates that a lower expected regret and a reduced rate of large regret cannot be achieved simultaneously.
* The authors have proposed an algorithm capable of achieving a Pareto trade-off between expected regret and tail rate.
* They have also extended these findings to a structured non-stationary bandit setting.


**Strengths:**

I find the authors' results compelling and their technical contributions to be of high value. This is indeed a noteworthy piece of work. The result provided could provide a general guideline (what can and cannot be acheived) for the MAB algorithm design.

**Weaknesses:**

NA

**Questions:**

* It would be beneficial to provide a clearer explanation of the intuition behind the "impossibility" result.
* Please elucidate the necessity for random permutation within the proposed algorithm.
* It would be interesting to explore the influence of the non-stationary constant $b_t$ on the performance of the algorithm. While Lemma 4.3 provides valuable insight, it would be helpful to understand why the result in Lemma 4.3 holds true despite $b_t$ potentially following any sequence (even one chosen in an adversarial manner).
* A numerical example illustrating the superior tail behavior of regret under the proposed algorithm, as compared to the conventional successive elimination algorithm, would be a beneficial addition.

---

> ### Author Rebuttal · Authors · 2023-08-09
>
> We sincerely thank you for your valuable time and review. We would like to provide responses to the four questions, which we find very helpful to improve our work.
>
> - Thanks for the question. In Table 1, we provide the critical values of the log tail probabilities, which serves as a more intuitive way to illustrate the impossibility results in Theorem 3.1 and Corollary 3.2. In the worst-case scenario, the log tail probability can be approximately regarded as $-x/T\cdot\text{(worst-case expected regret)}$, while in the instance-dependent scenario, the log tail probability can be approximately regarded as $-\text{(instance-dependent expected regret)}$. The intuition is that when the regret expectation becomes less optimal and consistent, more space is left for alleviating tail risks.
>
> - The randomization step is crucial to hedge against the baseline rewards $\{b_t\}$, for which deterministic algorithms may fail. The uniformly random permutation leads to two advantages that facilitate the analysis: (1) in each phase, the baseline rewards are transformed into "random noises''; (2) after each phase, the number of times any arm two arms in the active set have been pulled are the same. With uniform permutation, although the estimation of one arm is still biased, the difference between estimation for two different arms becomes unbiased.
>
> - As is suggested in the paragraph above, the randomization step is independent with the baseline rewards, and so it protects the policy against any potentially adversarial choice of $b_t$. In fact, the randomization idea is useful and necessary to hedge against an adversarial environment (see, e.g., [1]).
>
> - In the appendix, we provide a range of numerical experiments. We would like to emphasize that in both Figure 1 and 2, the first column indeed mimics the standard SE policy. This is because we set $\beta=0$ which makes $\text{rad}(n)$ dominated by the second term $\propto \sqrt{\ln T/n}$. One can observe that there is an experimental trade-off: Larger $\alpha$ (more sub-optimality) allows more light-tailed behavior on extreme values and larger $\beta$ (more inconsistency) gives rise to more concentration.
>
> [1] Krishnamurthy A, Wu ZS, Syrgkanis V (2018) Semiparametric contextual bandits. International Conference on Machine Learning, 2776–2785 (PMLR)

---

> > ### Comment · Reviewer_tbar · 2023-08-20
> >
> > Your responses addressed all my questions. I would like to keep my score.

---

### Decision · Program_Chairs · 2023-09-21

**Decision:**

Accept (spotlight)

**Comment:**

Though in a simple MAB setting, it's a good paper that proposes a policy that simultaneously juggles three balls: worst-case optimality, instance-dependent consistency, and light-tailed risk. The ML community could benefit from learning this work in an applied probability lens.